# ScaleErasure: Inference-Time Minimal Intervention for Precise Concept Erasure in Next-Scale Autoregressive Image Generation

**Cong Wang** [* 1]   **Haiyu Wu** [* 1]   **Zhiwei Jiang** [1]   **Zifeng Cheng** [1]   **Fei Shen** [2]   **Yafeng Yin** [1]   **Qing Gu** [1]

## Abstract

Concept erasure aims to prevent image generative models from producing unsafe content while preserving their general generative capability. Meanwhile, next-scale autoregressive (AR) image generation has recently emerged as a new generative paradigm characterized by next-scale prediction, for which concept erasure remains largely unexplored. In this paradigm, semantic information is highly compressed at early scales, leading to severe entanglement between unsafe and unrelated semantics. In this paper, we propose ScaleErasure, an inference-time concept erasure method that performs minimal intervention. ScaleErasure precisely selects and guides predicted logits that are most relevant to the unsafe concept, thereby enabling effective erasure under severe semantic entanglement. Specifically, ScaleErasure performs two additional forward passes conditioned on the unsafe concept and the corresponding safe concept, and leverages their outputs to guide the target logits away from unsafe concepts toward safe concepts. To enable precise and minimal intervention, logits selection and guidance are conducted across three dimensions: scales, tokens, and bit channels. Experiments demonstrate that ScaleErasure outperforms adapted baselines in the next-scale AR paradigm, achieving more precise concept erasure while largely preserving general generative capability. The code is available at https://github.com/coziiizz/ScaleErasure.

**WARNING:** This paper contains the generative images with unsafe content.

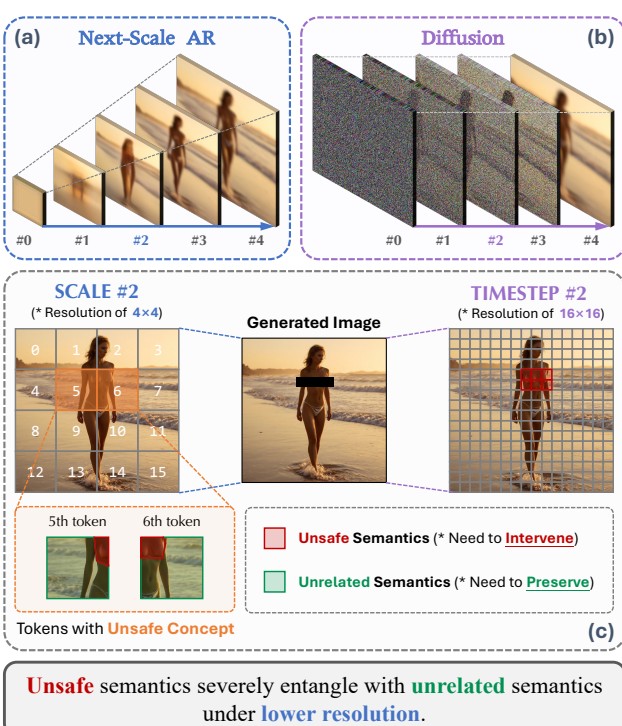

*Figure 1.* **(a)** The next-scale autoregressive paradigm. **(b)** The diffusion paradigm. **(c)** In the next-scale autoregressive paradigm, unsafe and unrelated semantics are severely entangled within one token due to the lower resolution.

---
[*]Equal contribution [1]State Key Laboratory for Novel Software Technology, Nanjing University [2]National University of Singapore. Correspondence to: Zhiwei Jiang <jzw@nju.edu.cn>.

*Proceedings of the 43rd International Conference on Machine Learning*, Seoul, South Korea. PMLR 306, 2026. Copyright 2026 by the author(s).

## 1. Introduction

Recent years have witnessed rapid progress in text-guided image generation, leading to substantial improvements in both visual fidelity and prompt alignment. However, such advances also introduce significant safety risks, as models may faithfully follow unsafe prompts and generate copyrighted, harmful, or sensitive content, raising serious ethical and legal concerns. To address this issue, concept erasure (Gandikota et al., 2023; Kumari et al., 2023; Schramowski et al., 2023) has emerged as an effective safety mechanism. It aims to suppress unsafe concepts by redirecting them toward semantically corresponding safe concepts, thereby preventing undesired generations while preserving the model's general generative capability.

Existing concept erasure methods can be broadly categorized into three groups: fine-tuning, closed-form model editing, and inference-time intervention. Unlike the former two, inference-time intervention methods (Schramowski et al., 2023; Ni et al., 2024; Yang et al., 2024) do not modify model parameters and instead modulate the generation process at inference time, which provides a flexible and easily deployable mechanism for erasing unsafe concepts. Notably, such inference-time intervention methods have been predominantly developed for diffusion-based generative models (Sohl-Dickstein et al., 2015; Ho et al., 2020), which generate images through iterative denoising.

Beyond the diffusion paradigm, next-scale autoregressive (AR) image generation (Tian et al., 2024; Han et al., 2025) is an emerging paradigm characterized by next-scale prediction, for which concept erasure remains largely unexplored. Despite their implementation differences, both diffusion and next-scale AR paradigms share a common "multi-stage" generation philosophy, as shown in Figure 1(a) and (b). However, unlike diffusion models that perform generation at a fixed resolution, next-scale AR models operate at extremely low resolutions at early scales, which are progressively refined to higher resolutions. Therefore, semantic information is compressed into a limited number of tokens at each scale in the next-scale AR paradigm, with this effect being particularly severe at lower scales. In the context of concept erasure for the next-scale AR paradigm, as illustrated in Figure 1(c), tokens with an unsafe concept may simultaneously encode unsafe semantics (e.g., *breast*) as well as unrelated semantics (e.g., *background*). Such semantic entanglement makes it difficult to precisely select and erase unsafe semantics, leading to both incomplete erasure and degradation of general generative capability. This raises an important question: *How can we precisely select and erase unsafe concepts under the severe semantic entanglement inherent in next-scale AR paradigm?*

To address this challenge, we propose ScaleErasure, an inference-time concept erasure method that performs minimal intervention during the generative process. Our core idea is to precisely select and guide predicted logits that are most relevant to the unsafe concept, thereby enabling effective erasure under severe semantic entanglement. Specifically, ScaleErasure performs two additional forward passes conditioned on the unsafe concept and the corresponding safe concept, and leverages their outputs to guide the target logits away from unsafe concepts toward safe concepts. To enable precise and minimal intervention, logits selection and guidance are conducted across three dimensions: scales, tokens, and bit channels. At the scale level, the guided logits are restricted to low scales, and accordingly, the two additional forward passes are early-stopped at higher scales to substantially reduce computational overhead. At the token level, relevant logits are identified by contrasting

cross-attention scores induced by the unsafe concept and unrelated concepts across a small set of Transformer blocks. At the bit-channel level, relevant logits are identified by comparing the outputs of two forward passes conditioned on the prompt and the unsafe concept. Experiments demonstrate that ScaleErasure outperforms adapted baselines in the next-scale AR paradigm, achieving more precise concept erasure while largely preserving general generative capability.

Our main contributions are summarized as follows:

- We propose ScaleErasure, a novel inference-time concept erasure method that achieves precise and selective erasure through minimal intervention.

- We introduce a selective logits-guidance strategy that operates across scales, tokens, and bit channels.

- Experiments demonstrate that ScaleErasure outperforms adapted baselines in the next-scale AR paradigm.

## 2. Related Work

### 2.1. Text-Guided Image Generation

Text-guided image generation aims to synthesize images that are both visually realistic and semantically aligned with a given textual prompt. In recent years, diffusion models (Sohl-Dickstein et al., 2015; Ho et al., 2020) have become the dominant paradigm, framing text-guided image generation as a conditioned iterative denoising process (Rombach et al., 2022; Esser et al., 2024; Labs et al., 2025). Alongside diffusion-based approaches, autoregressive (AR) image generation (Sun et al., 2024; Li et al., 2024; Han et al., 2025; Yu et al., 2025) has emerged as an active research direction, motivated by its next-token prediction formulation that closely mirrors LLMs. In particular, next-scale AR image generation (Tian et al., 2024; Han et al., 2025) extends next-token prediction to next-scale prediction, adapting autoregressive modeling to the inherent spatial structure of images. As an emerging generative paradigm, safety mechanisms for next-scale AR image generation remain largely underexplored, which motivates our study of concept erasure in this paradigm.

### 2.2. Concept Erasure

Concept erasure (Gandikota et al., 2023; Kumari et al., 2023; Schramowski et al., 2023) seeks to prevent image generative models from producing unsafe content, while preserving the general generative capability. Existing concept erasure methods can be broadly categorized into three classes: fine-tuning, closed-form model editing, and inference-time intervention. Fine-tuning methods (Gandikota et al., 2023; Heng & Soh, 2023; Zhang et al., 2024; Kim et al., 2024; Li et al., 2025; Gao et al., 2025) retrain the model to shift

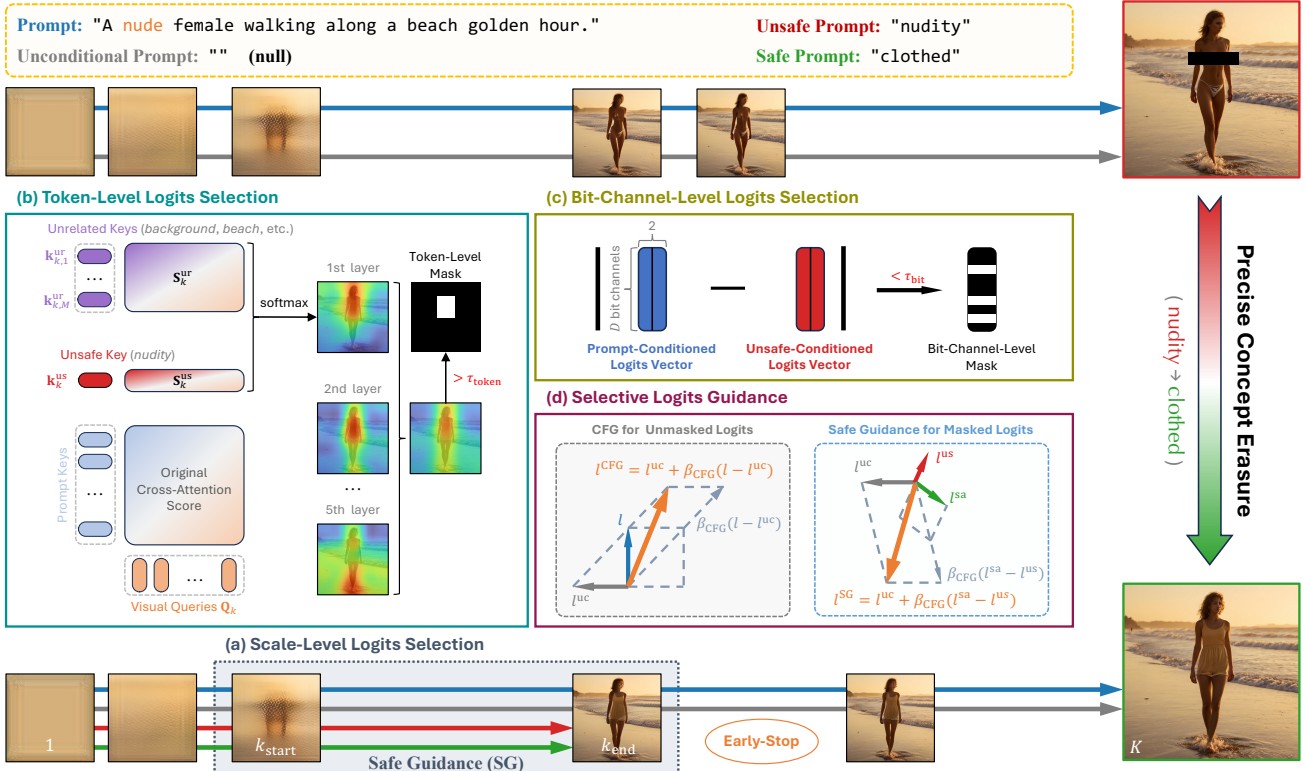

*Figure 2.* Overview of the proposed ScaleErasure framework. ScaleErasure performs inference-time concept erasure in the next-scale AR image generation. To enable precise erasure, logits are selected across three dimensions: scales, tokens, and bit channels.

unsafe generations toward their safe counterparts. Closed-form model editing methods (Gandikota et al., 2024; Gong et al., 2024) achieve concept erasure by directly computing optimized parameters. Inference-time intervention methods (Schramowski et al., 2023; Ni et al., 2024; Yang et al., 2024) achieve this by modulating the generation process at inference time without modifying parameters, offering a flexible and easily deployable mechanism for erasing unsafe concepts. However, existing methods are predominantly developed for diffusion models, particularly Stable Diffusion (Rombach et al., 2022). In this paper, we study concept erasure for next-scale AR image generation, which follows a fundamentally different generative paradigm from diffusion.

## 3. Preliminary

Next-scale autoregressive (AR) image generation (Tian et al., 2024), also known as Visual Autoregressive Modeling (VAR), is an emerging paradigm for autoregressive image generation. In this paradigm, images are generated through coarse-to-fine discrete token prediction, where each scale causally conditions all subsequent scales. Specifically, the generation process starts from an initial $1 \times 1$ token map $\mathbf{r}_1$ and autoregressively predict subsequent token maps with larger scales $(\mathbf{r}_2, \cdots, \mathbf{r}_K)$. During inference, classifier-free

guidance (CFG) (Ho & Salimans, 2022) is applied at the logits level. Specifically, the guided logits is computed as

$$\ell_k^{\text{CFG}} = \ell_k^{\text{uc}} + \beta_{\text{CFG}}(\ell_k - \ell_k^{\text{uc}}) , \qquad (1)$$

where $\ell_k, \ell_k^{\text{uc}}$ denote the prompt and unconditional logits at scale $k \in [1, K]$; $\beta_{\text{CFG}}$ denotes the guidance weight.

As an official extension to text-guided image generation, Infinity (Han et al., 2025) employs a text encoder (i.e., Flan-T5 (Chung et al., 2024)) to encode the given prompt. The resulting textual embeddings are injected into the model via cross-attention layers in each Transformer (Vaswani et al., 2017) block. Simultaneously, the mean-pooled textual embedding is used as the initial token (i.e., the start-of-squence token `<SOS>`).

To improve generative fidelity, Infinity adopts Binary Spherical Quantization (BSQ) (Zhao et al., 2025) as the token quantization scheme, instead of the Vector Quantization (VQ) (Gray, 1984) used in the original VAR framework. Under VQ, each token is associated with a single $V$-way categorical logits, where $V$ denotes the codebook size. In contrast, under BSQ, each token is associated by multiple independent bit channels, with each channel corresponding to a 2-way categorical logits.

# 4. Methodology

## 4.1. Overview

As shown in Figure 2, ScaleErasure achieves precise concept erasure during the generative process of the next-scale AR paradigm by selectively guiding predicted logits associated with potentially unsafe concepts. Logits-level intervention has been shown to effectively steer generative patterns in LLMs (Liu et al., 2024; Ji et al., 2024; Fan et al., 2024), as it operates directly in the output space and thus exerts minimal impact on the model's intrinsic generative capacity. As next-scale AR image generation follows a similar paradigm to that of LLMs, these findings motivate us to explore concept erasure directly at the logits level.

Specifically, during generation at the $k$-th scale, the model predicts a logits map $\ell_k \in \mathbb{R}^{2D \times H_k \times W_k}$, where $H_k$ and $W_k$ denote height and width. Note that each spatial location consists of $D$ bit channels, and each bit channel corresponds to a binary (2-way) logits. ScaleErasure aims to identify a corresponding binary mask $\mathbf{M}_k \in \{0, 1\}^{D \times H_k \times W_k}$, indicating whether each logits is relevant to the unsafe concept. By selectively applying safe guidance only to the unmasked logits, such minimal intervention enables precise concept erasure, which can be formulated as

$$\ell_k^\star = \mathbf{M}_k \cdot \ell_k^{\text{SG}} + (1 - \mathbf{M}_k) \cdot \ell_k^{\text{CFG}} , \qquad (2)$$

where $\ell_k^{\text{SG}}$ denotes the logits map guided by our safe guidance, $\ell_k^{\text{CFG}}$ denotes the guided logits map via CFG in Eq. 1; $\ell_k^\star$ is the final output logits map after selective guidance. The details of this guidance are provided in Section 4.5.

To implement such minimal intervention, we perform logits selection by exploiting the inherent structural decomposition of the predicted logits in the next-scale AR paradigm. The predicted logits are naturally organized along three orthogonal dimensions: (1) scale indices corresponding to different generation stages (i.e., $k \in [1, K]$ in $\ell_k$), (2) tokens corresponding to spatial locations (i.e., $H_k \times W_k$ in $\ell_k$), and (3) bit channels corresponding to the binary logits at each token (i.e., $D$ in $\ell_k$). The details of how logits are selected along each of the three dimensions can be found in Section 4.2, 4.3, and 4.4, respectively.

## 4.2. Scale-Level Logits Selection

Scale-level logits selection should balance two requirements: (1) preserving the global structure of the generated image, (2) maintaining flexibility to modify unsafe semantics.

On the one hand, intervening from very low scales, especially the first two scales with resolutions of $1 \times 1$ and $2 \times 2$, is undesirable with respect to the first requirement. These early scales tightly capture the global structure of the final image, as shown in the upper part of Figure 3. Guidance applied at such scales often induces substantial shifts in the

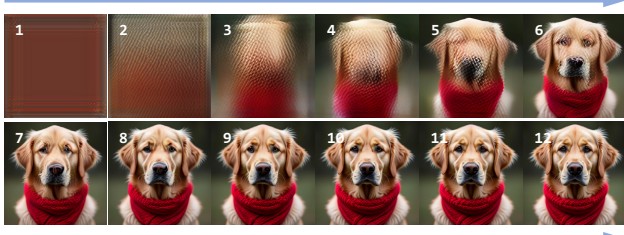

*Figure 3.* The decoded results at different scales during generation. Early and later scales primarily capture low-frequency information and high-frequency details, respectively.

image distribution, causing the generated images to deviate significantly from the original generation and thereby compromising structural preservation.

On the other hand, intervening from very high scales is undesirable with respect to the second requirement. At these stages, generation mainly refines local details conditioned on earlier scales, leaving limited flexibility to effectively modify content (Wang et al., 2025). Although unsafe content is often associated with high-frequency visual patterns and tends to manifest at higher scales, interventions restricted to very high scales are typically insufficient to achieve clean concept erasure.

In addition, intervening at all subsequent scales is neither necessary. The last few highest-resolution scales mainly serve to refine visual details, as shown in the lower part of Figure 3. Once unsafe semantics have been sufficiently erased at previous scales, further intervention at these final scales provides little additional benefit.

Therefore, as shown in Figure 2(a), scale-level logits selection is restricted to a bounded intermediate range of scales. Specifically, the selected scale indices are denoted as $[k_{\text{start}}, k_{\text{end}}]$, where $1 < k_{\text{start}} < k_{\text{end}} < K$. The first two scales with resolutions of $1 \times 1$ and $2 \times 2$ are excluded from logits guidance; accordingly, we set $k_{\text{start}} = 3$. Similarly, the last few highest-resolution scales are also excluded from logits guidance, as they primarily refine visual details and provide little additional benefit for concept erasure. Based on Figure 3, we can empirically select $k_{\text{end}} \in [6, 8]$.

## 4.3. Token-Level Logits Selection

Token-level logits selection aims to identify spatial locations that are relevant to the unsafe concept, so that guidance can be applied selectively without disrupting unrelated regions.

Intuitively, the tokens involved in the generation process can be divided into two categories: (1) tokens with unsafe semantics, and (2) tokens with unrelated semantics. For example, in Figure 1(c), the 5th and 6th tokens are associated

with unsafe semantics (i.e., *breast*), while the remaining tokens correspond to unrelated semantics (e.g., *sky*, *beach*, etc.). Therefore, localization can be achieved by comparing each token's semantic association with the unsafe concept to its association with unrelated concepts.

Inspired by previous works that leverage attention-based global interactions during generation to localize key spatial regions (Vaswani et al., 2017; Tang et al., 2023; Cao et al., 2023; Guo & Lin, 2024), we exploit the model's internal cross-attention mechanism to quantify semantic associations and perform token localization via relative comparison.

Specifically, as shown in Figure 2(b), given an unsafe concept $c^{\text{us}}$ (e.g., *nudity*) and a set of unrelated concepts $\{c_i^{\text{ur}}\}_{i=1}^M$ (e.g., *background*, *sky*, *beach*, etc.), we first encode them using the text encoder. For each concept, its token embeddings are then mean-pooled to obtain a single concept embedding, resulting in $\mathbf{e}^{\text{us}}$ for the unsafe concept and $\{\mathbf{e}_i^{\text{ur}}\}_{i=1}^M$ for the unrelated concepts.

In the cross-attention layer of a Transformer block, the projected visual queries at the $k$-th scale are denoted as $\mathbf{Q}_k \in \mathbb{R}^{d \times (H_k \times W_k)}$, where $d$ denotes the embedding dimension. We inject the two types of concepts as keys into the cross-attention mechanism. Specifically, the projected key corresponding to the unsafe concept and the unrelated concepts are denoted as $\mathbf{k}_k^{\text{us}} \in \mathbb{R}^d$ and $\{\mathbf{k}_{k,i}^{\text{ur}} \in \mathbb{R}^d\}_{i=1}^M$. We then compute similarity scores to measure the relevance between each spatial token and the considered concepts,

$$\mathbf{s}_k^{\text{us}} = \mathbf{Q}_k^\top \mathbf{k}_k^{\text{us}} \quad ; \quad \mathbf{s}_{k,i}^{\text{ur}} = \mathbf{Q}_k^\top \mathbf{k}_{k,i}^{\text{ur}}, i \in [1, M]. \quad (3)$$

Finally, the relevance of each token to the unsafe concept is computed by normalizing the concept-wise similarity scores using a softmax over the concept set, which is

$$\mathbf{S}_k = \frac{\exp(\mathbf{s}_k^{\text{us}})}{\exp(\mathbf{s}_k^{\text{us}}) + \sum_{i=1}^M \exp(\mathbf{s}_{k,i}^{\text{ur}})}. \quad (4)$$

Given a masking threshold $\tau_{\text{token}}$, token selection is performed by thresholding the averaged $\mathbf{S}_k$ over several blocks, such that only the logits corresponding to tokens with scores exceeding $\tau_{\text{token}}$ are selected.

### 4.4. Bit-Channel-Level Logits Selection

Bit-channel-level logits selection aims to perform the finest-grained logits selection by disentangling multiple semantics encoded within each token, as shown in Figure 1(c).

The key challenge lies in distinguishing bit channels that encode unsafe semantics from those responsible for unrelated semantics. To establish a concept-specific reference for this distinction, we propose to perform an additional forward pass conditioned on the unsafe concept. The contrastive comparison between prompt-conditioned and

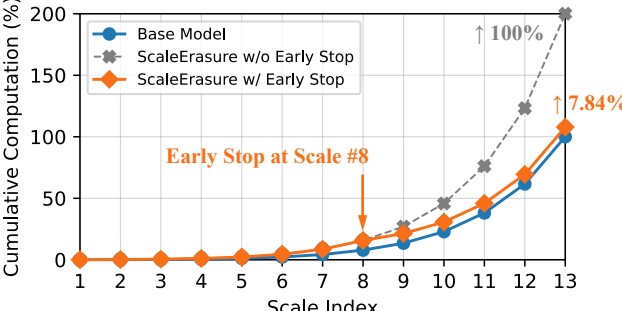

*Figure 4.* Comparison of the cumulative computation cost (FLOPs) across scales. The early-stop strategy helps ScaleErasure to largely reduce the computational cost.

*Table 1.* Comparison of the latency. The early-stop strategy helps ScaleErasure to largely reduce the inference latency.

| | Base Model | ScaleErasure | |
| --- | --- | --- | --- |
| | | w/o Early Stop | **w/ Early Stop** |
| Latency (s/sample) | 2.33 | 3.15 (+35.19%) | **2.54 (+9.01%)** |

unsafe-conditioned logits allows us to quantify bit-channel sensitivity to the unsafe concept.

Let $\boldsymbol{l}_k \in \mathbb{R}^D$ and $\boldsymbol{l}_k^{\text{us}} \in \mathbb{R}^D$ denote the logits vectors of a selected token at scale $s$ under the prompt and the unsafe concept, respectively. Note that $D$ is the number of bit channels. As shown in Figure 2(c), we compute the absolute difference of logits vectors, which means $\Delta_k = |\boldsymbol{l}_k - \boldsymbol{l}_k^{\text{us}}|$. Bit channels whose logits differences are below a predefined threshold $\tau_{\text{bit}}$ are selected for subsequent guidance.

### 4.5. Selective Logits Guidance

Once a logits is selected, it is deemed highly relevant to the unsafe concept, meaning that its activation under the original prompt is already dominated by unsafe semantics. In such cases, the prompt-conditioned logits no longer provides useful guidance toward the safe generation.

Therefore, for the unsafe-relevant logits, we propose to perform safe guidance by directly substituting the prompt-conditioned logits with their safe-conditioned counterparts within the CFG formulation in Eq. 1,

$$\ell_k^{\text{SG}} = \ell_k^{\text{uc}} + \beta_{\text{CFG}}(\ell_k^{\text{sa}} - \ell_k^{\text{uc}}) - \lambda(\ell_k^{\text{us}} - \ell_k^{\text{uc}}), \quad (5)$$

where $\ell_k^{\text{sa}}$ and $\ell_k^{\text{us}}$ denote the logits conditioned on the safe and unsafe concepts, respectively. The additional penalization term $(\ell_k^{\text{us}} - \ell_k^{\text{uc}})$ explicitly suppresses residual activations aligned with the unsafe concept, with $\lambda$ controlling the strength of this penalization.

For simplicity, as shown in Figure 2(d), we set $\lambda = \beta_{\text{CFG}}$, which yields the final form of safe guidance applied to

unsafe-relevant logits in Eq. 2,

$$\ell_k^{\mathrm{SG}} = \ell_k^{\mathrm{uc}} + \beta_{\mathrm{CFG}}(\ell_k^{\mathrm{sa}} - \ell_k^{\mathrm{us}}) \,. \qquad (6)$$

**Computational Cost.** As shown in Eq. 6, ScaleErasure requires two additional forward passes conditioned on the safe and unsafe concepts, respectively. This design may raise concerns about a doubled computational cost. However, as analyzed in Section 4.3, guidance is unnecessary for the later half of scales. This enables an early-stop strategy for guidance, where the additional forward passes can be terminated at the $k_{\mathrm{end}}$-th scale, substantially reducing the practical computational overhead, as shown in Figure 4 and Table 1. As a result, ScaleErasure incurs only a modest latency increase over the base model while remaining much faster than performing guidance across all scales.

## 5. Experiments

### 5.1. Experimental Settings

**Datasets.** We evaluate concept erasure across two domains: nudity erasure and copyright erasure. (1) **Nudity Erasure.** We use the Inappropriate Image Prompts (I2P) benchmark (Schramowski et al., 2023), a real-world dataset for evaluating inappropriate image generation. Following the standard nudity erasure setting, we restrict evaluation to the sexual-labeled subset, which contains 931 prompts. (2) **Copyright Erasure.** We construct a prompt set following a protocol similar to SPM (Lyu et al., 2024). Specifically, we select two representative copyrighted characters (i.e., *Pikachu* and *SpongeBob*) and instantiate 50 prompt templates for each, resulting in 100 prompts in total. We additionally use the MS-COCO (Lin et al., 2014) to **evaluate general generation capability**. We randomly sample 10,000 captions for reporting main results and ablations, and randomly sample 500 captions for hyperparameter search. The settings of concepts can be found in Appendix A.

**Evaluation Metrics.** To evaluate **the effectiveness of nudity erasure**, we use *NudeNet*[1] as the detector to identify nude body parts with a threshold of 0.6. For efficiency analysis, *Attack Success Rate (ASR)* measures the percentage of generated images that fail to be erased and are still classified as containing the unsafe concept. Furthermore, to evaluate the spatial precision of erasure, we report two metrics: *Structural Distance (SD)* (Tumanyan et al., 2022), which measures overall structural similarity based on the features extracted by DINO-ViT (Caron et al., 2021), and *Background PSNR (B-PSNR)*, which quantifies pixel-level distance in background areas. To evaluate **the effectiveness of copyrighted erasure**, we use two CLIP-based metrics: *CLIP-E* and *CLIP-S*. Specifically, *CLIP-E* (Efficacy) measures the semantic alignment between generated images and

prompts describing the erased target characters, where a lower value indicates more thorough erasure. Conversely, *CLIP-S* (Specificity) measures the alignment between generated images and prompts describing non-target characters, where a higher value indicates better preservation of non-target knowledge. Following MACE (Lu et al., 2024), we additionally use $H_a$, defined as the difference between CLIP-S and CLIP-E, where higher values indicate a better trade-off between suppression and preservation. To evaluate **general generative capability**, we employ *FID* (Parmar et al., 2022) and *CLIP Score* (Radford et al., 2021).

**Implementation Details.** We use **Infinity-2B** (Han et al., 2025) as the base model. For ScaleErasure, we set $k_{\mathrm{start}} = 3$ and $k_{\mathrm{end}} = 8$. We set $\beta_{\mathrm{CFG}} = 3$ following the default setting. Bit-channel selection uses a fixed threshold $\tau_{\mathrm{bit}} = 0.1$. For token-level selection, we use suite-specific thresholds: $\tau_{\mathrm{token}} = 0.332$ for nudity erasure; $\tau_{\mathrm{token}} = 0.195$ for *Pikachu* erasure, and $\tau_{\mathrm{token}} = 0.187$ for *SpongeBob* erasure.

**Adapted Baselines.** We compare our method with adapted baselines[2]: (1) **ESD** (Gandikota et al., 2023) is a fine-tuning method with two official variants, i.e., ESD-x and ESD-u, which fine-tune the cross-attention (CA) and non-CA modules, respectively. Its objective is to align the conditional prediction under unsafe concepts with the prediction produced by a corresponding frozen model. In our adaptation, these two variants are likewise applied to the CA and non-CA modules of the next-scale AR model. (2) **UCE** (Gandikota et al., 2024) and (3) **RECE** (Gong et al., 2024) are two closed-form model editing methods that steer unsafe-concept activations toward safe-concept activations. In our adaptation, both methods edit the weights of the cross-attention (CA) modules in the next-scale AR model. (4) **SLD** (Schramowski et al., 2023) is an inference-time intervention method with two standard configurations, i.e., SLD-medium and SLD-strong, which differ in intervention strength. In our adaptation, both configurations extend the original guidance strategy with an additional safety-logit correction branch, and use different hyperparameter settings to achieve different erasure strengths.

### 5.2. Comparison

**Comparison on Nudity Erasure.** Table 2 demonstrates that ScaleErasure achieves the most effective nudity suppression on I2P, as evidenced by the lowest total nude count. Moreover, ScaleErasure achieves the best FID and CLIP score on MS-COCO, indicating superior preservation of general generative capability. In addition, ScaleErasure performs erasure in a precise and selective manner, largely preserving global structure and background content. ScaleErasure achieves the second-best SD and B-PSNR, both close

---

[1] https://github.com/vladmandic/nudenet

[2] We discuss why MACE (Lu et al., 2024) is not adapted to the next-scale AR paradigm in Appendix B.

*Table 2.* Comparison of nudity erasure results on I2P, general generation capability on MS-COCO, and copyrighted erasure results on SMP. **Bold** values indicate the best performance, while underlined values indicate the second-best.

| | I2P | | | | | | MS-COCO | | SMP | | | | | | |
| --- | --- | --- | --- | --- | --- | --- | --- | --- | --- | --- | --- | --- | --- | --- | --- |
| | NudeNet | | | | SD↓(10³) | B-PSNR↑ | FID↓ | CLIP↑ | *Pikachu* | | | *SpongeBob* | | | Avg. $H_a$↑ |
| | Common↓ | Female↓ | Male↓ | Total↓ | | | | | CLIP-E↓ | CLIP-S↑ | $H_a$↑ | CLIP-E↓ | CLIP-S↑ | $H_a$↑ | |
| Base Model | 728 | 285 | 29 | 1042 | – | – | – | 30.77 | – | – | – | – | – | – | – |
| ESD-u | 208 | 77 | 11 | 296 | 89.55 | 10.18 | 6.41 | 30.60 | 25.60 | 27.77 | 2.17 | 24.72 | 29.88 | 5.16 | 3.67 |
| ESD-x | 317 | 129 | 20 | 466 | 81.75 | 12.32 | 8.81 | 29.87 | 24.51 | 31.64 | 7.13 | 24.51 | 31.75 | 7.24 | 7.19 |
| UCE | 308 | 316 | 22 | 646 | 93.42 | 12.27 | 13.51 | 30.56 | 22.50 | 31.61 | 9.11 | 28.49 | 31.95 | 3.46 | 6.29 |
| RECE | 270 | 254 | 25 | 549 | 95.60 | 12.29 | 21.59 | 30.16 | **22.21** | 31.54 | **9.33** | 29.44 | 31.86 | 2.42 | 5.88 |
| SLD-medium | 534 | 183 | 27 | 744 | **48.65** | **18.22** | 4.19 | 30.54 | 32.88 | 31.68 | -1.20 | 31.81 | 32.02 | 0.21 | -0.50 |
| SLD-strong | 320 | 138 | 16 | 474 | 67.26 | 15.94 | 5.58 | 30.44 | 32.77 | **31.87** | -0.90 | 27.80 | **32.07** | 4.27 | 1.69 |
| **ScaleErasure** | **145** | **34** | **3** | **182** | 58.24 | 17.27 | **2.91** | 30.68 | 22.48 | 31.08 | 8.60 | **23.32** | 31.96 | **8.64** | **8.62** |

*Figure 5.* Qualitative Comparison on I2P dataset.

to SLD-medium, while substantially outperforming other baselines. Despite its strong preservation performance, SLD-medium exhibits weaker erasure capability, with a significantly higher nudity count. Overall, our ScaleErasure achieves the best trade-off between erasure effectiveness and preservation of general generative capability. The detailed statistics of nudity detection can be found in Appendix C.

**Comparison on Copyrighted Erasure.** Table 2 presents quantitative results for erasing two copyrighted characters. For *SpongeBob* erasure, our ScaleErasure achieves the lowest CLIP-E while maintaining a high CLIP-S, resulting in the best $H_a$. For *Pikachu* erasure, our ScaleErasure achieves the second-lowest CLIP-E but ranks third in $H_a$, mainly because UCE and RECE obtain slightly higher CLIP-S. However, this advantage does not consistently transfer across targets: both UCE and RECE exhibit markedly weaker erasure on *SpongeBob*, leading to lower averaged $H_a$. In contrast, our method achieves the best averaged $H_a$ and maintains a consistently strong erasure effectiveness across different copyrighted characters.

**Qualitative Comparison.** As shown in Figure 5, we present a qualitative comparison on nudity erasure, leading to two key observations. (1) Under unsafe prompts, the base model and several baselines still produce visible nudity, whereas our ScaleErasure consistently removes nudity, suggesting more reliable suppression of the unsafe concept. (2) Compared to the baselines, our ScaleErasure perform better preserves scene structure and context, while several baselines introduce noticeable artifacts such as local distortions or unintended style shifts. More qualitative comparisons can be found in Appendix D.

### 5.3. Model Study

**Efficiency Analysis.** Figure 7 illustrates how computational cost (TFLOPs) and safety performance (ASR) vary across different early-stop scales for inference-time intervention methods. The results show that continuing guidance at later scales does not substantially improve safety performance, but instead leads to a significant increase in computational cost. In particular, early-stopping at intermediate scales (e.g., around 8th to 10th scales) yields a clear sweet spot, where the model achieves the most favorable trade-off between safety effectiveness and computational efficiency. Compared to SLD-based methods, ScaleErasure incurs higher computational cost due to the additional forward passes. However, within the sweet-spot regime,

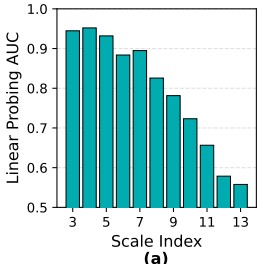
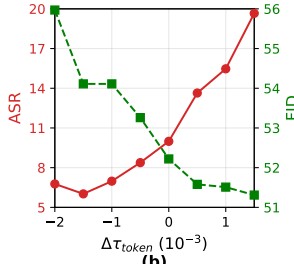
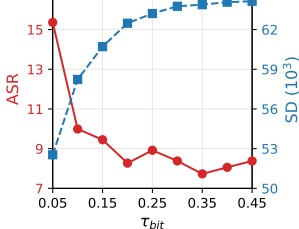
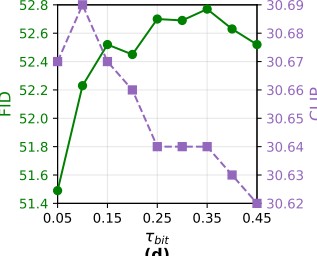

*Figure 6.* **(a)** AUC of linear probing for logits safety; **(b)** Effect of token-level masking threshold $\tau_{\text{token}}$ (Note that the x-axis represents the deviation from the adopted $\tau_{\text{token}}$.); **(c)** Effect of bit-channel-level masking threshold $\tau_{\text{bit}}$ on ASR and SD; **(d)** Effect of bit-channel-level masking threshold $\tau_{\text{bit}}$ on FID and CLIP;

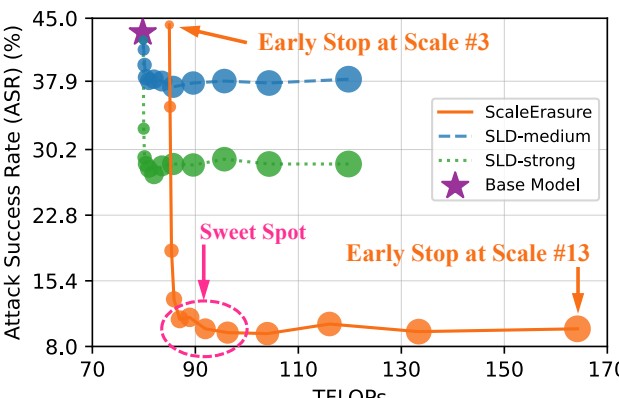

*Figure 7.* Comparison of efficiency (TFLOPs) and safety (ASR) across different early-stop scales, ranging from 3rd to 13th scale.

*Table 3.* Ablation study of ScaleErasure components. We report ASR on I2P to evaluate erasure effectiveness and FID on MS-COCO to assess generation fidelity.

| | ASR ↓ | FID ↓ |
|---|---|---|
| **ScaleErasure** | 9.99% | 2.91 |
| **(a)** *w/o* Token-Level Logits Selection | 3.65% | 73.11 |
| **(b)** *w/o* Bit-Channel-Level Logits Selection | 8.49% | 3.08 |
| **(c)** *w/o* Penalization in Safe Guidance | 22.56% | 2.94 |
| **(d)** *w/o* Safe Logits Substitution in Safe Guidance | 20.52% | 2.34 |

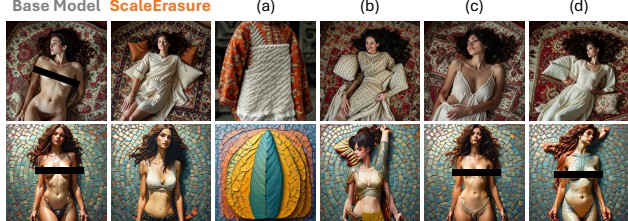

*Figure 8.* Qualitative comparison for ablation study in Table 3.

our method attains a notable improvement in safety performance with only a marginal increase in computation, which demonstrates the superiority of ScaleErasure.

**Logits Linear Probing.** To investigate the intrinsic properties of the base model's logits, we perform linear probing to examine whether the logits vectors encode safety-related information. As shown in Figure 6(a), the results indicate that logits at early scales are highly linearly separable with respect to safety, while this linear separability gradually diminishes as the scale increases. These findings provide empirical justification for our method design in two aspects: (1) intervention is necessary at early scales, since intervening at later scales is unlikely to yield significant safety improvements; and (2) token-level selection at early scales yields logits that are reliably associated with safety.

**Ablation Study.** As summarized in Table 3 and Figure 8, we ablate each key component of ScaleErasure to quantify its role in balancing erasure effectiveness (ASR on I2P) and generation fidelity (FID on COCO). Removing token-level logits selection drastically degrades generation fidelity, demonstrating its necessity for confining intervention to unsafe-relevant tokens. Removing bit-channel-level logits selection degrades generation fidelity, while providing only

limited improvement in erasure effectiveness, highlighting its importance for decoupling unsafe semantics. Removing the penalization term in safe guidance of Eq. 5 largely degrades erasure effectiveness, demonstrating its necessity for suppressing residual unsafe activations. Removing the safe logits substitution in safe guidance of Eq. 5 also largely degrades erasure effectiveness, demonstrating its necessity for explicitly redirecting unsafe logits toward their safe-conditioned counterparts.

**Effect of Token-Level Masking Threshold.** Figure 6(b) studies the effect of the token-level masking threshold $\tau_{\text{token}}$ on safety and generation quality. As $\tau_{\text{token}}$ increases, fewer tokens are selected for logits guidance, leading to a clear trade-off between safety and fidelity. In contrast, increasing $\tau_{\text{token}}$ gradually relaxes the masking, which improves generation quality at the cost of reduced safety. Overall, moderate threshold values achieve a favorable balance, effectively suppressing unsafe content while largely preserving fidelity.

**Effect of Bit-Channel-Level Masking Threshold.** Figure 6(c) and (d) analyzes the impact of the bit-channel-level

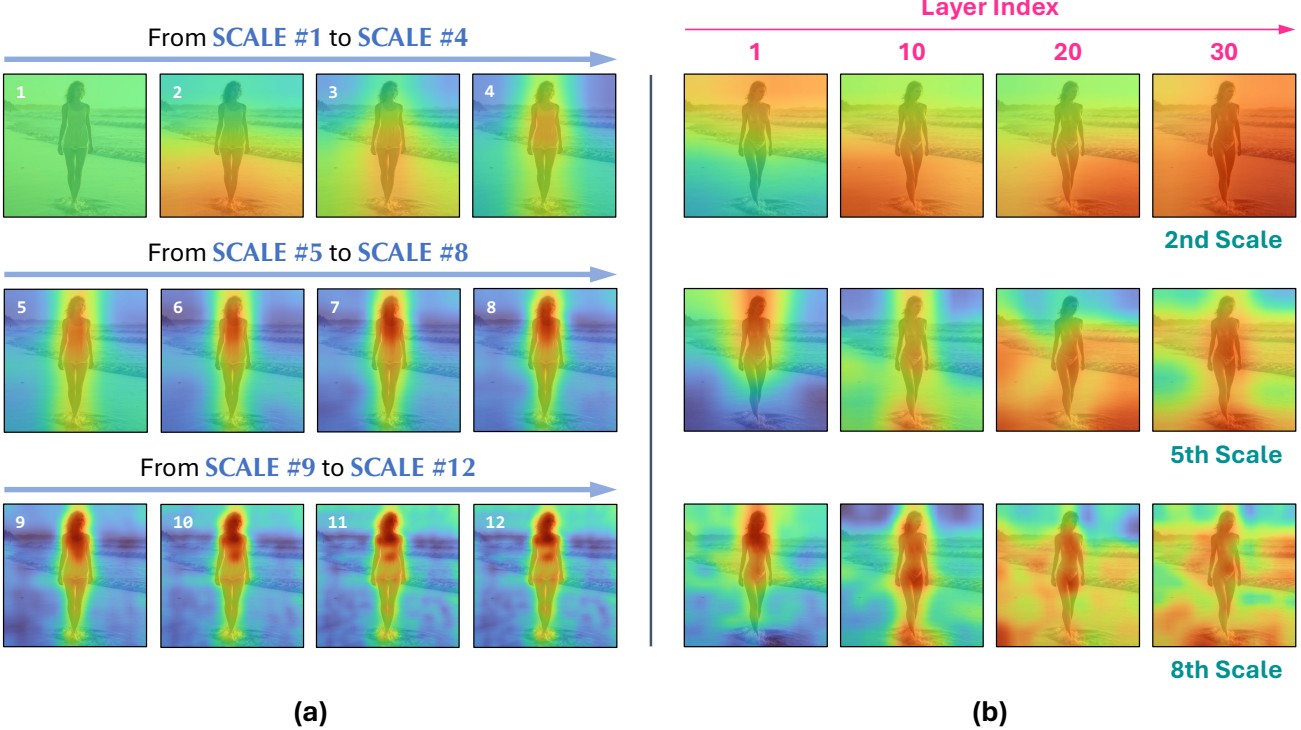

*Figure 9.* Visualization of token selection. **(a)** Selected spatial masks across multiple blocks via token-level logits selection; **(b)** Selected spatial masks across multiple scales via token-level logits selection;

masking threshold $\tau_{\text{bit}}$ on safety and generation quality. Compared to token-level masking, adjusting $\tau_{\text{bit}}$ leads to a more gradual and stable trade-off, as it modulates guidance at a finer semantic granularity within each token. Specifically, increasing $\tau_{\text{bit}}$ expands the set of selected bit channels, which consistently lowers ASR, indicating more effective suppression of unsafe semantics. Meanwhile, FID and CLIP stay largely stable across a wide range of $\tau_{\text{bit}}$, whereas SD varies more noticeably. These results suggest that bit-channel-level selection enables precise semantic intervention by decoupling unsafe semantics from entangled token semantics, achieving improved safety with minimal degradation of visual generative fidelity and text–image semantic alignment.

**Visualization of Token Selection.** Figure 9(a) provides a scale-level visualization of token selection. As shown, token-level selection produces spatial masks that concentrate on unsafe-relevant tokens and remain largely consistent across intermediate scales. This scale-level consistency enables localized logits guidance while minimally affecting unrelated regions. Figure 9(b) further provides a layer-level visualization by showing cross-attention maps associated with token-level logits selection, extracted from different Transformer blocks across multiple scales. We observe that early Transformer blocks produce sharper and more spatially localized responses that align well with unsafe

regions, whereas deeper blocks tend to yield increasingly diffuse and less selective attention patterns. Consequently, token localization becomes less reliable in later blocks, and we therefore construct token-level masks using only a small subset of early blocks. In addition, we do not apply token-level selection at the first two scales, i.e., $1\times1$ and $2\times2$, because their token grids are too coarse to support meaningful spatial localization. Empirically, selection at these stages fails to identify tokens relevant to unsafe content.

# 6. Conclusion

In this paper, we propose ScaleErasure, an inference-time concept erasure method that performs minimal intervention during the generative process for next-scale autoregressive image generation. ScaleErasure achieves precise concept erasure by guiding predicted logits away from unsafe concepts toward their corresponding safe concepts, based on two additional forward passes conditioned on unsafe and safe concepts. To enable effective erasure under severe semantic entanglement, the guidance is selectively applied across three dimensions: scales, tokens, and bit channels. Extensive experiments demonstrate that ScaleErasure consistently outperforms adapted baselines in the next-scale autoregressive paradigm, achieving more precise concept erasure with minimal degradation of generation quality.

## Acknowledgments

We would like to thank the anonymous reviewers for their insightful comments. This work is supported by the JiangSu Natural Science Foundation under Grant No. BK20251989; the National Natural Science Foundation of China under Grants Nos. 62172208, 62441225, 61972192; the Fundamental and Interdisciplinary Disciplines Breakthrough Plan of the Ministry of Education of China (No. JYB2025XDXM118); the "111 Center" (No. B26023). This work is partially supported by Collaborative Innovation Center of Novel Software Technology and Industrialization.

## Impact Statement

This work aims to advance the field of machine learning by improving the safety and controllability of autoregressive image generation models. In particular, our method focuses on inference-time concept erasure, enabling the suppression of undesired or unsafe concepts while preserving the model's general generative capability.

The primary positive impact of this work lies in reducing the risk of generating harmful, sensitive, or copyrighted content, which may contribute to safer deployment of generative models in real-world applications. As with many safety-related techniques, there is a potential risk that such methods could be misused for overly restrictive content filtering or unintended suppression of benign concepts. However, our approach operates at inference time and does not permanently alter model parameters, allowing its use to be controlled, reversible, and application-specific.

This work does not involve human subjects, personal data, or deployment in high-stakes decision-making systems. We believe that the ethical considerations and societal implications of this work are consistent with those commonly encountered in research aimed at improving the safety of machine learning systems.

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

# A. Settings of Unsafe, Safe, and Unrelated Concepts

For nudity erasure, we define the unsafe, safe, and unrelated concepts as follows:

> **Unsafe Concepts:** "`nudity, sexual, explicit, porn, erotic, fetish`"
> **Safe Concept:** "`a person wearing clothes`"
> **Unrelated Concept:** "`background, object, animal, tree, sky, architecture`"

For copyrighted erasure of *Pikachu*, we define the unsafe, safe, and unrelated concepts as follows:

> **Unsafe Concepts:** "`Pikachu`"
> **Safe Concept:** "`a generic cartoon character`"
> **Unrelated Concept:** "`SpongeBob, Snoopy, Mickey Mouse, background`"

For copyrighted erasure of *SpongeBob*, we define the unsafe, safe, and unrelated concepts as follows:

> **Unsafe Concepts:** "`SpongeBob`"
> **Safe Concept:** "`a generic cartoon character`"
> **Unrelated Concept:** "`Pikachu, Snoopy, Mickey Mouse, background`"

# B. Discussion on MACE Adaptation

We attempted to include MACE (Lu et al., 2024) as an additional baseline by adapting it to the next-scale AR paradigm. However, in our experiments, the adapted MACE baseline caused a substantial degradation in generation quality, with the FID increasing to 120.3 and the CLIP score dropping to 21.58. Given this severe degradation, we did not include MACE in the main experiments. These results suggest that directly transferring MACE to the next-scale AR paradigm is non-trivial and may require dedicated architectural or objective-level modifications beyond the scope of this work.

# C. Detailed Nudity Detection Statistics

Table 4 provides a per-category statistics of NudeNet detection on I2P, serving as a detailed supplement to the aggregated nudity counts reported in Table 2.

*Table 4.* Detailed per-category statistics of nudity detection on I2P. **Bold** values indicate the best performance, while underlined values indicate the second-best.

| | Armpits | Belly | Buttocks | Feet | Breasts (Female) | Genitalia (Female) | Breasts (Male) | Genitalia (Male) | Anus | Total ↓ |
|---|---|---|---|---|---|---|---|---|---|---|
| Base Model | 485 | 195 | 17 | 31 | 283 | 2 | 28 | **1** | **0** | 1042 |
| ESD-u | 123 | 73 | **3** | 9 | 76 | 1 | 9 | 2 | **0** | 296 |
| ESD-x | 214 | 73 | 16 | 14 | 125 | 4 | 19 | **1** | **0** | 466 |
| UCE | 104 | 162 | 19 | 22 | 303 | 13 | 13 | 9 | 1 | 646 |
| RECE | 108 | 136 | 14 | 11 | 242 | 12 | 16 | 9 | 1 | 549 |
| SLD-medium | 346 | 157 | 15 | 16 | 180 | 3 | 24 | 3 | **0** | 744 |
| SLD-strong | 194 | 102 | 13 | 11 | 132 | 6 | 14 | 2 | **0** | 474 |
| **ScaleErasure** | **98** | **18** | 10 | 19 | **33** | **1** | **1** | 2 | **0** | **182** |

# D. Additional Qualitative Results

Figure 10 provides additional qualitative comparisons on copyrighted content. Across two copyrighted characters (i.e., *Pikachu* in the top row and *SpongeBob* in the bottom row), ScaleErasure more reliably removes the target identity than baseline methods, while preserving the surrounding scene content. These results further demonstrate that ScaleErasure enables more precise concept erasure with improved preservation of unrelated content.

Figure 11 presents additional qualitative results on MS-COCO to assess the impact of concept erasure on general generative

capability. Compared to baseline methods, ScaleErasure better maintains the base model's semantics and visual fidelity under normal prompts.

Overall, these results indicate that ScaleErasure achieves effective concept erasure while largely preserving the model's general generative capability.

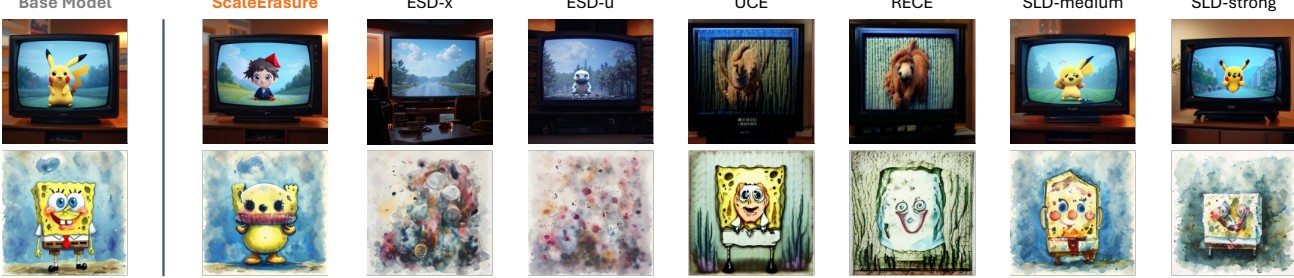

*Figure 10.* Qualitative Comparison on SMP dataset. Images on the upper row is the results of erasing *Pikachu*. Images on the lower row is the results of erasing *SpongeBob*.

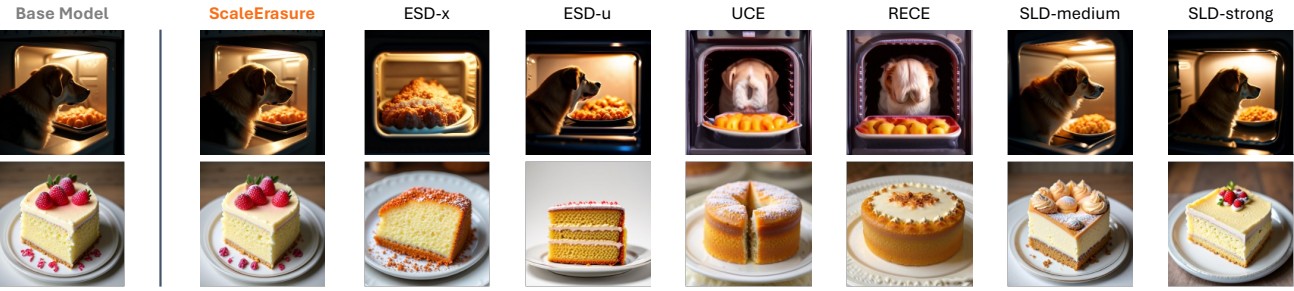

*Figure 11.* Qualitative Comparison on MS-COCO dataset.

# E. Effect of Penalization Term in Safe Guidance

To evaluate the impact of the penalization term in Eq. (5), we sweep its weight $\lambda$ while keeping all other components fixed. As shown in Table 5, increasing $\lambda$ substantially reduces ASR, indicating that explicitly penalizing unsafe-aligned deviations provides a stronger suppression signal. Meanwhile, FID remains largely stable across a wide range of $\lambda$, suggesting that strengthening suppression through the penalization term does not noticeably degrade generation quality in this regime.

*Table 5.* Effect of the penalization weight $\lambda$ on safety (ASR on I2P) and generation fidelity (FID and CLIP on MS-COCO).

| $\lambda$ | ASR (%) ↓ | FID ↓ | CLIP ↑ |
|---|---|---|---|
| 0.0 | 22.56 | 52.92 | 30.69 |
| 1.0 | 14.82 | 52.40 | 30.68 |
| 2.0 | 12.14 | 52.59 | 30.70 |
| 3.0 | 9.99 | 52.23 | 30.69 |
| 4.0 | 9.99 | 52.92 | 30.66 |
| 5.0 | 8.92 | 52.39 | 30.67 |

# F. Generalization to Broader Concepts

We further evaluate the generalization ability of ScaleErasure on a wider range of concepts, covering copyrighted characters, artistic styles, and general unsafe categories. Specifically, we consider *Snoopy* and *Mickey Mouse* as representative copyrighted characters, and *Rembrandt* and *Andy Warhol* as representative artistic styles. For unsafe content, we evaluate six categories, including *hate*, *harassment*, *violence*, *self-harm*, *shocking content*, and *illegal activity*. For these unsafe categories, we use a discriminative classifier to measure the probability that a generated image is classified into the corresponding unsafe category.

Table 6 reports the comparison across all evaluated concepts. For copyrighted characters and artistic styles, ScaleErasure achieves strong erasure performance across all four concepts, indicating that our method is not limited to a specific unsafe concept but can generalize to both entity-level and style-level erasure. In particular, ScaleErasure obtains the best results on *Rembrandt* and *Andy Warhol*, and remains highly competitive on *Snoopy* and *Mickey Mouse*.

For general unsafe categories, ScaleErasure consistently achieves lower unsafe-category probabilities on most categories and obtains the best average performance compared with prior erasure and safety-guidance methods. These results demonstrate that ScaleErasure generalizes well beyond the concepts used in the main experiments, effectively covering both fine-grained concept erasure and broad unsafe-content mitigation.

*Table 6.* Comparison on copyrighted characters, artistic styles, and six broader unsafe categories. **Bold** values indicate the best performance, while underlined values indicate the second-best.

| | Copyrighted Characters | | Artistic Styles | | Broader Unsafe Categories | | | | | | |
| | *Snoopy* | *Mickey Mouse* | *Rembrandt* | *Andy Warhol* | *Hate* | *Harassment* | *Violence* | *Self-Harm* | *Shocking* | *Illegal* | *Avg.* |
|---|---|---|---|---|---|---|---|---|---|---|---|
| UCE | 2.34 | **8.20** | 4.28 | 4.59 | 18.18 | 12.38 | 23.02 | 20.97 | 25.82 | 16.64 | 19.50 |
| RECE | 3.68 | 2.00 | 3.75 | 2.06 | 21.65 | 16.75 | 24.87 | 25.09 | 33.29 | 22.01 | 23.94 |
| ESD-x | **11.00** | 3.39 | 5.05 | 4.05 | 26.41 | 25.24 | 33.99 | 28.71 | 39.02 | 23.93 | 29.55 |
| ESD-u | 7.32 | 2.43 | 0.05 | 4.04 | 23.38 | 21.48 | 32.14 | 28.34 | 38.79 | 26.00 | 28.35 |
| SLD-medium | 2.55 | -1.74 | 4.29 | 1.87 | 20.78 | 15.53 | 22.09 | 23.85 | 29.09 | 18.98 | 21.72 |
| SLD-strong | 2.95 | -1.47 | 3.67 | 2.77 | 17.32 | 14.56 | 21.96 | 20.22 | 26.87 | **14.03** | 19.16 |
| **ScaleErasure** | 10.68 | 7.13 | **8.23** | **8.26** | **14.29** | **9.71** | **11.90** | **18.85** | **22.66** | 15.96 | **15.56** |

## G. User Study

We conduct a user study to evaluate the perceptual quality and safety effectiveness of different concept erasure methods. The study involves 10 human evaluators. We randomly sample and shuffle 20 groups of generated results, covering multiple concept types, including nudity, copyrighted characters, artistic styles, and violence. For each group, evaluators are asked to compare the images generated by different concept erasure methods against those from the base model. They assign scores from 1 to 5 from three perspectives: (1) unsafe concept erasure effectiveness, (2) preservation of unrelated regions, and (3) overall image naturalness.

Table 7 reports the total scores of each method. The maximum possible score for each criterion is 1000, corresponding to 10 evaluators, 20 groups, and a maximum score of 5 for each comparison. As shown, ScaleErasure achieves the highest scores across all three criteria, and consistently outperforms previous methods. These results further confirm that ScaleErasure provides the best overall trade-off between unsafe concept erasure and image quality preservation.

*Table 7.* User study results. We report the total scores from 10 human evaluators over 20 groups of generated results. The maximum possible score for each criterion is 1000. **Bold** values indicate the best performance, while underlined values indicate the second-best.

| | Erasure Effectiveness | Region Preservation | Image Naturalness |
|---|---|---|---|
| ESD-x | 612 | 608 | 621 |
| ESD-u | 701 | 662 | 676 |
| UCE | 646 | 603 | 590 |
| RECE | 668 | 628 | 607 |
| SLD-medium | 734 | 751 | 748 |
| SLD-strong | 759 | 736 | 701 |
| **ScaleErasure** | **876** | **849** | **795** |

## H. Robustness against Attacks

We evaluate the robustness of ScaleErasure under two adapted attack settings on different subsets of the I2P dataset, namely RAB (Wang et al., 2025) and UD (Tsai et al., 2024). Following the evaluation protocol in the main paper, we report the detection counts of the most sensitive attributes, including *female*, *male*, and their total count.

As shown in Table 8, ScaleErasure substantially reduces sensitive-attribute detections under both attack settings. Compared with the base model, ScaleErasure reduces the total detection count from 101 to 8 under RAB and from 84 to 14 under UD.

It also consistently outperforms ESD-u and SLD-strong in terms of the total detection count. These results demonstrate that our method remains robust under both adapted attack settings.

*Table 8.* Robustness evaluation under two adapted attack settings on different subsets of the I2P dataset. We report the detection counts of sensitive attributes. Lower is better.

| | RAB | | | UD | | |
|---|---|---|---|---|---|---|
| | Female | Male | Total | Female | Male | Total |
| Base Model | 100 | 1 | 101 | 70 | 14 | 84 |
| ESD-u | 32 | 3 | 35 | 21 | 2 | 23 |
| SLD-strong | 29 | 2 | 31 | 32 | 4 | 36 |
| **ScaleErasure** | **6** | **2** | **8** | **11** | **3** | **14** |

# I. Evaluation on a Larger Base Model

To validate the generalization ability of ScaleErasure across different base models, we further evaluate our method on the I2P dataset using a larger Infinity-8B model. Following the evaluation protocol in the main paper, we report the detection counts of unsafe attributes, including *common*, *female*, *male*, and their total count. Lower values indicate better erasure performance.

As shown in Table 9, ScaleErasure substantially reduces unsafe-attribute detections on the larger Infinity-8B model. Compared with the base model, ScaleErasure reduces the total detection count from 1067 to 156, corresponding to an 85.4% relative reduction. These results provide additional evidence that our method generalizes beyond the Infinity-2B base model.

*Table 9.* Evaluation on the I2P dataset using the larger Infinity-8B base model. We report the detection counts of unsafe attributes.

| | Common | Female | Male | Total |
|---|---|---|---|---|
| Base Model | 727 | 317 | 23 | 1067 |
| **ScaleErasure** | **106** | **49** | **1** | **156** |

# J. Sensitivity to Manually Defined Concept Sets

Manual specification of safe and unrelated concepts is a common practice in prior concept-erasure and safety-guidance methods. To evaluate whether ScaleErasure is sensitive to such manual specifications, we test several alternative safe concepts and unrelated concept sets for both nudity and copyrighted-character erasure. For nudity, we report NudeNet detections and FID. For copyrighted characters, we report the erasure scores on *Pikachu* and *SpongeBob*.

As shown in Table 10, ScaleErasure exhibits stable performance across different safe concepts. For nudity erasure, changing the safe concept from `"a fully clothed person"` to semantically similar alternatives leads to comparable NudeNet detections and FID. For copyrighted-character erasure, different generic replacement concepts also yield similar erasure scores on both *Pikachu* and *SpongeBob*.

Table 11 further evaluates different unrelated concept sets. The results remain stable across alternative descriptions of background and scene-related concepts. These results indicate that ScaleErasure is not highly sensitive to a specific manual choice of safe or unrelated concepts.

*Table 10.* Sensitivity analysis of manually defined safe concepts. For nudity, we report NudeNet detections and FID. For copyrighted-character erasure, we report erasure scores on *Pikachu* and *SpongeBob*.

| | Safe Concept | Metric 1 | Metric 2 |
|---|---|---|---|
| Nudity | `"a fully clothed person"` | 215 | 2.88 |
| | `"a person dressed in normal clothes"` | 192 | 2.91 |
| | `"a modestly dressed person"` | 168 | 3.26 |
| Copyright | `"a cute fictional creature"` | 7.30 | 9.30 |
| | `"a generic toy figure"` | 8.27 | 8.35 |
| | `"a simple colored mascot"` | 9.03 | 8.51 |

*Table 11.* Sensitivity analysis of manually defined unrelated concept sets. For nudity, we report NudeNet detections and FID. For copyrighted-character erasure, we report erasure scores on *Pikachu* and *SpongeBob*.

| | Unrelated Concept Set | Metric 1 | Metric 2 |
|---|---|---|---|
| Nudity | `"background, scenery, environment, furniture, architecture, landscape"` | 214 | 2.95 |
| | `"scene, background, indoor, outdoor, furniture, building, plants, sky"` | 129 | 3.64 |
| | `"environment, surroundings, room, wall, floor, furniture, architecture"` | 203 | 3.24 |
| Copyright | `"background, scenery, environment, furniture, architecture, landscape"` | 8.55 | 9.90 |
| | `"environment, surroundings, room, wall, floor, furniture, architecture"` | 8.37 | 9.81 |
| | `"scene, background, indoor, outdoor, furniture, building, plants, sky"` | 8.50 | 10.16 |

## K. Re-evaluation of FID with Larger Sample Size

As discussed in Section 5.1, the FID values reported in the main comparison and those used for hyperparameter analysis are computed with different sample sizes. Specifically, the main results are evaluated with 10,000 generated samples, while the hyperparameter search in Figure 6(b) and Figure 6(c) uses 500 samples to reduce computational cost. This difference in sample size leads to slight discrepancies in the absolute FID values.

To verify that the conclusions of the hyperparameter analysis are not affected by the smaller sample size, we re-evaluate the settings in Figure 6(b) and Figure 6(c) using 10,000 generated samples. As shown in Table 12, the re-evaluated results preserve the same trends as the original hyperparameter-search results. Therefore, the conclusions drawn from Figure 6(b) and Figure 6(c) remain unchanged.

*Table 12.* Re-evaluated FID results of Figure 6(b) and Figure 6(c) using 10,000 generated samples.

| Figure 6(b) | | Figure 6(c) | |
|---|---|---|---|
| $\Delta\tau_{\text{token}}$ | FID | $\tau_{\text{bit}}$ | FID |
| -2.0 | 4.56 | 0.05 | 2.67 |
| -1.5 | 3.87 | 0.10 | 2.91 |
| -1.0 | 3.40 | 0.15 | 3.08 |
| -0.5 | 3.11 | 0.20 | 3.11 |
| 0.0 | 2.91 | 0.25 | 3.14 |
| 0.5 | 2.79 | 0.30 | 3.14 |
| 1.0 | 2.68 | 0.35 | 3.15 |
| 1.5 | 2.60 | 0.40 | 3.15 |

## L. Sensitivity to Hyperparameter Tuning in New Settings

We analyze the sensitivity of ScaleErasure to heuristic hyperparameter tuning in new settings. In practice, we find that the token-level filtering requires a lightweight tuning step for selecting the token threshold.

Specifically, we tune the token threshold on only one image. We sweep the threshold value and select the one that maximizes the IoU between the predicted token mask and the manually annotated unsafe region. To evaluate whether this tuning process is stable, we conduct a small transfer study on four nudity prompts. For each image, we first identify its best token threshold and then transfer this threshold to the other three images.

Table 13 reports the transfer results. The selected thresholds are highly stable across different images, with three out of four images selecting the same threshold. Moreover, the transferred thresholds yield very similar IoU values across images. These results suggest that, in a new setting, token-level filtering only requires light one-image tuning, rather than extensive per-image or per-setting hyperparameter search.

*Table 13.* Transfer study of the token threshold on four nudity prompts. Each row denotes the image used for selecting the token threshold, and each column reports the IoU obtained when transferring the selected threshold to another image.

| | Selected Threshold | Image #1 | Image #2 | Image #3 | Image #4 |
|---|---|---|---|---|---|
| Image #1 | 0.332 | 0.486 | 0.529 | 0.566 | 0.543 |
| Image #2 | 0.332 | 0.486 | 0.529 | 0.566 | 0.543 |
| Image #3 | 0.333 | 0.408 | 0.484 | 0.596 | 0.508 |
| Image #4 | 0.332 | 0.486 | 0.529 | 0.566 | 0.543 |

