# OpenReview forum: "ScaleErasure: Inference-Time Minimal Intervention for Precise Concept Erasure in Next-Scale Autoregressive Image Generation"
_ICML.cc/2026/Conference — ICML 2026 regular_

### Official Review · Reviewer_Amjf · 2026-03-11

**Soundness:** 3
**Presentation:** 3
**Significance:** 3
**Originality:** 3
**Overall Recommendation:** 5
**Confidence:** 3

**Summary:**

The paper propose ScaleErasure, an inference-time method that selectively modifies predicted logits to suppress unsafe concepts. Unlike parameter-editing approaches, the method performs minimal intervention during generation by guiding logit away from unsafe concepts toward corresponding safe concepts. It use three levels of selective intervention: scale-level selection, token-level selection, bit-channel-level selection. The method requires two additional forward passes conditioned on safe and unsafe concept. Experiment on nudity and copyrighted character removal show improved erasure effectiveness while maintaining generation quality compared with adapted baselines such as ESD, UCE, RECE, and SLD.

**Compliance With Llm Reviewing Policy:**

Affirmed.

**Final Justification:**

Thank you to the authors for the rebuttal. My concerns have been addressed, so I have raised my rating.

**Key Questions For Authors:**

If the concerns are addressed, I would be willing to further raise the rating to 5 or 6.

**Limitations:**

Discussing the limitations or failure cases would further strengthen the work.

**Strengths And Weaknesses:**

### Strengths
1. The paper address concept erasure in next-scale AR image generation, which is relatively unexplored compared to diffusion-based models.
2. The three-level selection strategy (scale, token, bit-channel) is well aligned with the structure of AR generation and enable precise intervention.
3. Experiment across multiple benchmarks demonstrate that the proposed method achieves a strong trade-off between safety and generation quality.

### Weakness
1. The approach require two additional forward passes, increasing inference cost compared to simpler intervention methods.
2. Experiment only cover nudity and two copyrighted characters, leaving open questions about generalization to other unsafe concepts, like violence, celebrities, or styles.
3. The paper compares against several representative baselines (ESD, UCE, RECE, and SLD). However, it would strengthen the empirical evaluation if the authors could also include MACE as a baseline.
4. Incorporating experiments against attack methods like UD [1], RAB [2], etc would strengthen the work.
5. The author could improve the paper's comprehensiveness by incorporating a discussion of related work, ANT [3] and EraseAnything [4].
6. Discussing the limitations or failure cases would further strengthen the work.

[1] To generate or not? safety-driven unlearned diffusion models are still easy to generate unsafe images... for now

[2] Ring-a-bell! how reliable are concept removal methods for diffusion models?

[3] Set You Straight: Auto-Steering Denoising Trajectories to Sidestep Unwanted Concepts

[4] Eraseanything: Enabling concept erasure in rectified flow transformers

---

> ### Author Rebuttal · Authors · 2026-03-31
>
> Dear reviewer Amjf:
>
> **We sincerely appreciate your recognition of the timeliness, the method design, and the strong experimental results. We will address your concerns below.**
>
> ---
>
> **`W1:` Extra inference cost from additional forward passes**
>
> **`Response:`**
>
> We agree that our method incurs additional cost due to the extra forward passes. However, this overhead remains modest in practice: with the early-exit strategy, the inference cost increases by only 7.84% (Figure 4). More importantly, this limited overhead yields substantially better performance than the baselines, leading to a much more favorable safety-efficiency trade-off (Figure 6).
>
> ---
>
> **`W2:` Limited generalization evaluation across concepts**
>
> **`Response:`**
>
> Thanks for your insightful suggestion. We conduct experiments on a broader range of concepts, including **Snoopy** and **Mickey Mouse** for copyrighted characters, **Rembrandt** and **Andy Warhol** for art styles, as well as six broader unsafe categories. Specifically, for the six broader unsafe categories, we use a discriminative classifier to measure the probability that a generated image belongs to the corresponding unsafe category.
>
> |$H_a$|Snoopy|Mickey Mouse|Rembrandt|Andy Warhol|
> |-|:-:|:-:|:-:|:-:|
> |UCE|2.34|**8.20**|4.28|4.59|
> |RECE|3.68|2.00|3.75|2.06|
> |ESD-x|**11.00**|3.39|5.05|4.05|
> |ESD-u|7.32|2.43|0.05|4.04|
> |SLD-medium|2.55|-1.74|4.29|1.87|
> |SLD-strong|2.95|-1.47|3.67|2.77|
> |**ScaleErasure (Ours)**|10.68|7.13|**8.23**|**8.26**|
>
> | |Hate|Harassment|Violence|Self-Harm|Shocking|Illegal|*Avg.*|
> |-|:-:|:-:|:-:|:-:|:-:|:-:|:-:|
> |ESD-x|18.18|12.38|23.02|20.97|25.82|16.64|19.5|
> |ESD-u|21.65|16.75|24.87|25.09|33.29|22.01|23.94|
> |UCE|26.41|25.24|33.99|28.71|39.02|23.93|29.55|
> |RECE|23.38|21.48|32.14|28.34|38.79|26.00|28.35|
> |SLD-medium|20.78|15.53|22.09|23.85|29.09|18.98|21.72|
> |SLD-strong|17.32|14.56|21.96|20.22|26.87|**14.03**|19.16|
> |**ScaleErasure (Ours)**|**14.29**|**9.71**|**11.90**|**18.85**|**22.66**|15.96|**15.56**|
>
> These results show that **ScaleErasure generalizes well to a broader range of concepts**. We will include the detailed results in the revised version.
>
> ---
>
> **`W3:` Missing a baseline of MACE**
>
> **`Response:`**
>
> Thanks for your helpful suggestion. We attempted to include MACE as a baseline by adapting it to the next-scale AR paradigm. However, in our experiments, the adapted MACE baseline led to a substantial degradation in generation quality, with **FID increasing to 120.3** and **CLIP score dropping to 21.58**. Given the severe degradation, we did not include it in the main experiments. We will clarify this observation in the revised version.
>
> ---
>
> **`W4:` Missing robustness evaluation against attacks**
>
> **`Response:`**
>
> Thanks for your helpful suggestion. We conduct robustness evaluations under two adapted attack settings on different subsets of the I2P dataset, namely **(1) RAB** and **(2) UD**, and report the detection counts of the most sensitive attributes.
>
> | |Female (1)|Male (1)|Total (1)|Female (2)|Male (2)|Total (2)|
> |:-|:-:|:-:|:-:|:-:|:-:|:-:|
> |Base Model|100|**1**|101|70|14|84|
> |ESD-u|32|3|35|21|**2**|23|
> |SLD-strong|29|2|31|32|4|36|
> |**ScaleErasure (Ours)**|**6**|2|**8**|**11**|3|**14**|
>
> These results show that our method remains robust under both attack settings. We will include the detailed results in the revised version.
>
>
> ---
>
> **`W5:` Missing discussion of ANT and EraseAnything in the section of Related Work**
>
> **`Response:`**
>
> Thanks for your helpful suggestion. In the revised version, we will include these two fine-tuning-based methods in Related Work.
>
> > ANT steers denoising trajectories away from unwanted concepts without relying on heuristic anchor selection. EraseAnything formulates concept erasure as a bi-level optimization problem with LoRA-based tuning and attention regularization.
>
> ---
>
> **`W6:` Lack of limitation/failure case discussion**
>
> **`Response:`**
>
> Thanks for your helpful suggestion. In the revised version, we will clarify the limitation of limited automation, since the concepts still needs to be manually specified. We will also discuss a representative failure case in multi-subject scenes, where precise erasure of only the target-related subject becomes more challenging.
>
> ---
>
> **Lastly, we sincerely thank you again for the detailed and constructive comments, which will help us improve the paper. We are happy to clarify any further questions during the discussion period.**

---

> > ### Author Rebuttal · Reviewer_Amjf · 2026-04-04
> >
> > Thank you to the authors for the rebuttal. My concerns have been addressed, so I have raised my rating.

---

> > > ### Author Response · Authors · 2026-04-04
> > >
> > > Dear reviewer Amjf:
> > >
> > > Thank you sincerely for taking the time to review our rebuttal and for thoughtfully considering our clarifications. We truly appreciate your increased score and your constructive comments, which help us strengthen the quality and clarity of our work.
> > >
> > > Best regards,
> > >
> > > The Authors

---

### Official Review · Reviewer_bSLj · 2026-03-13

**Soundness:** 3
**Presentation:** 4
**Significance:** 3
**Originality:** 3
**Overall Recommendation:** 4
**Confidence:** 4

**Summary:**

This paper introduces ScaleErasure, the first framework for concept erasure in next-scale autoregressive (AR) image models. To address semantic entanglement across scales, it proposes a fine-grained intervention mechanism operating on scale, token, and bit-channel dimensions. By leveraging parallel safe/unsafe guidance, the method dynamically suppresses harmful features during inference without retraining. Experiments on Infinity-2B demonstrate effective removal of unsafe concepts (e.g., nudity) while preserving high image quality and semantic consistency.

**Compliance With Llm Reviewing Policy:**

Affirmed.

**Key Questions For Authors:**

Please refer to the weakness.

**Limitations:**

Yes.

**Strengths And Weaknesses:**

**Strengths**
1. The paper is well-written with intuitive figures and diagrams that effectively communicate the motivation behind "semantic entanglement" and the core contributions of the proposed method.
2. ScaleErasure is a timely contribution as the first framework to tackle concept erasure specifically for next-scale autoregressive (AR) image models. The qualitative results are impressive, demonstrating successful removal of unsafe concepts while maintaining high visual fidelity.
3. The experimental setup is solid, featuring a good balance of quantitative metrics, qualitative comparisons, and efficiency analysis. Furthermore, the ablation studies provide convincing evidence for the necessity and effectiveness of each proposed module.

**Weakness**
1. The current experiments are primarily based on Infinity-2B. It would be helpful to include additional results on other base models (e.g., different scales or architectures) to demonstrate the generalizability of the proposed method better.

2. While FLOPs are reported, it would be more intuitive to see actual generation latency comparisons (e.g., seconds per image) under common configurations—with and without early stopping—compared to the base model. This would give readers a clearer sense of the practical overhead and the real-world benefit of the early-stop strategy.

3. Regarding the token-level masking threshold (τ_token), the paper shows that different concepts may require relatively fine-tuned threshold values. Could the authors clarify the selection logic? Is there a suggestion range that works reasonably well across concepts, or does it typically require per-concept tuning? Some practical guidance here would be helpful for real-world deployment.

---

> ### Author Rebuttal · Authors · 2026-03-31
>
> Dear reviewer bSLj:
>
> **We sincerely appreciate your recognition of the motivation and the experiments of our work. We will address your concerns below.**
>
> ---
>
> **`W1:` Lack of validation on diverse base models.**
>
> **`Response:`**
>
> Thanks for your helpful suggestion. We further evaluate our method on the I2P dataset using a larger Infinity-8B model. The results show that ScaleErasure remains highly effective in suppressing unsafe content. This provides additional evidence that our method generalizes beyond the Infinity-2B base model. We will include the detailed results in the revised version.
>
> | |Common|Female|Male|Total|
> |-|-|-|-|-|
> |Base Model|727|317|23|1067|
> |**ScaleErasure  (Ours)**|**106**|**49**|**1**|**156**|
>
> ---
>
> **`W2:` Missing latency comparison**
>
> **`Response:`**
>
> Thanks for your insightful suggestion. We report the generation latency on an NVIDIA RTX A6000, which shows that the early-stop strategy effectively reduces the overhead and that the runtime increase of our method remains acceptable in practice.
>
> | |Latency (s/sample)|
> |-|:-:|
> |*Base Model*| 2.33|
> |ScaleErasure (w/o early stop)|3.15 (+35.19%)|
> |**ScaleErasure (w/ early stop)**|**2.54 (+9.01%)**|
>
> ---
>
> **`W3:` Clarification on token-level masking threshold selection**
>
> **`Response:`**
>
> Thank you for pointing this out. The token-level filtering does require a simple tuning step.
> Specifically, we tune $\tau_\mathrm{token}$ on **only one image** by sweeping it and selecting the value that maximizes IoU between the predicted token mask and the manually annotated unsafe region.
>
> To verify whether this tuning is stable, we conduct a small transfer study on four nudity prompts. For each image, we first identify its best threshold and then transfer it to the other three images. The selected threshold is highly stable across images. Moreover, the transferred thresholds yield very similar IoU values across images. These results suggest that token-level filtering requires only light one-image tuning in practice, rather than extensive per-image or per-setting hyperparameter search.
>
> | |Select Token Threshold|#1|#2|#3|#4|
> |-|:-:|:-:|:-:|:-:|:-:|
> |#1|0.332|0.486|0.529|0.566|0.543|
> |#2|0.332|0.486|0.529|0.566|0.543|
> |#3|0.333|0.408|0.484|0.596|0.508|
> |#4|0.332|0.486|0.529|0.566|0.543|
>
> ---
>
> **Lastly, we sincerely thank you again for the detailed and constructive comments, which will help us improve the paper. We are happy to clarify any further questions during the discussion period.**

---

> > ### Author Rebuttal · Reviewer_bSLj · 2026-04-04
> >
> > Thanks for the rebuttal. My concerns have been fully resolved.

---

> > > ### Author Response · Authors · 2026-04-04
> > >
> > > Dear reviewer bSLj:
> > >
> > > Thank you sincerely for taking the time to review our rebuttal and for thoughtfully considering our clarifications. We truly appreciate your positive score and your constructive comments, which help us strengthen the quality and clarity of our work.
> > >
> > > Best regards,
> > >
> > > The Authors

---

### Official Review · Reviewer_KhEd · 2026-03-16

**Soundness:** 2
**Presentation:** 3
**Significance:** 2
**Originality:** 2
**Overall Recommendation:** 4
**Confidence:** 3

**Summary:**

This paper studies concept erasure for next-scale autoregressive image generation models. The authors propose ScaleErasure, an inference-time method that performs selective guidance at three levels: scale, token, and bit-channel. The approach is evaluated on nudity removal and copyrighted character erasure. Experimental results show improved erasure success while preserving overall image quality.

**Compliance With Llm Reviewing Policy:**

Affirmed.

**Final Justification:**

Most of my concerns have been addressed. Please include latency measurements, as FLOP calculations alone may not accurately reflect actual runtime performance.

**Key Questions For Authors:**

Please refer to the weakness I list.

**Limitations:**

No.

**Strengths And Weaknesses:**

**Strength**

1. The paper is clearly written and easy to follow.

2. The proposed method operates entirely during inference which makes it potentially easier to deploy in practice.

**Weakness**

1. Instead of performing erasure during the generation process, why not leverage a language model to determine whether the prompt is safe? If the prompt is unsafe, the model could simply rewrite it to remove the unsafe concepts. This approach appears to be more straightforward and potentially more effective, as it addresses the root cause of the issue.

2. Key components of the method rely on fixed hyperparameters or heuristics (e.g., scale selection ranges and threshold choices for token- and bit-channel filtering). This raises the concern of whether the method requires extensive hyperparameter tuning when applied to new settings.

3. It is unclear whether the unsafe words are manually marked or detected automatically by a model. In either case, the unsafe words must be explicitly identified to enable the feedforward process. This further reinforces my first concern: if unsafe words are already being detected, why not simply rewrite the prompt to remove them or refuse to generate the corresponding content?

4. The method requires additional forward passes. There is not a detailed runtime comparison with SLD and other inference-time methods.

The first one is my biggest concern.

---

> ### Author Rebuttal · Authors · 2026-03-31
>
> Dear reviewer KhEd:
>
> **We sincerely appreciate your recognition of the paper presentation and the method design. We will address your concerns below.**
>
> ---
>
> **`W1:` Why not use prompt rewriting**
>
> **`Response:`**
>
> Thanks for your insightful comment. Prompt rewriting [1] is another research direction that operates at a different stage of the safety pipeline. In general, generative safety can be addressed at three stages: (1) pre-generation rewriting, (2) in-generation intervention, and (3) post-generation filtering. Our work focuses on the generation-time stage rather than prompt-level preprocessing.
>
> While prompt rewriting is straightforward, **it cannot address unsafe generative dynamics originating from the model’s internal knowledge, which may be activated by adversarial prompts** [2]. Therefore, generation-time control remains necessary to dynamically steer the process based on the evolving visual content, and thus cannot be fully replaced by prompt rewriting.
>
> [1] Ores: Open-vocabulary responsible visual synthesis. AAAI 2023
>
> [2] MMA-Diffusion: MultiModal Attack on Diffusion Models, CVPR 2024
>
> ---
>
> **`W2:` Potential sensitivity to heuristic hyperparameter tuning in new settings**
>
> **`Response:`**
>
> Thank you for pointing this out. The scale selection range and the bit-channel filtering threshold generalize well across new settings. In contrast, token-level filtering does require a simple tuning step.
>
> Specifically, we tune $\tau_\mathrm{token}$ on **only one image** by sweeping it and selecting the value that maximizes IoU between the predicted token mask and the manually annotated unsafe region.
>
> To verify whether this tuning is stable, we conduct a small transfer study on four nudity prompts. For each image, we first identify its best threshold and then transfer it to the other three images. The selected threshold is highly stable across images. Moreover, the transferred thresholds yield very similar IoU values across images. These results suggest that, in a new setting, token-level filtering requires only light one-image tuning rather than extensive per-image or per-setting hyperparameter search.
>
> | |Select Token Threshold|#1|#2|#3|#4|
> |-|:-:|:-:|:-:|:-:|:-:|
> |#1|0.332|0.486|0.529|0.566|0.543|
> |#2|0.332|0.486|0.529|0.566|0.543|
> |#3|0.333|0.408|0.484|0.596|0.508|
> |#4|0.332|0.486|0.529|0.566|0.543|
>
> ---
>
> **`W3:` Unclear whether unsafe words are manually defined or automatically detected**
>
> **`Response:`**
>
> In the setting of concept erasure, the unsafe words are manually defined for the unsafe concept. These unsafe words are not required to explicitly appear in the prompt. Instead, they serve as concept descriptions that specify the erasure target.
>
> This is also different from prompt rewriting or refusal. Prompt rewriting operates only on the input text, whereas our method focuses on generation-time intervention. Even when unsafe words do not appear explicitly in the prompt, the model may still generate unsafe content due to its internal knowledge and generative dynamics, especially under adversarial prompting. Therefore, the ability to specify an unsafe concept does not make prompt rewriting sufficient, since the core problem here is **how to suppress that concept during generation** rather than merely edit the input prompt.
>
> ---
>
> **`W4:` Lack of detailed runtime comparison with the inference-time baselines**
>
> **`Response:`**
>
> In Figure 6, we compared the runtime efficiency of our method with inference-time baselines (i.e., SLD-medium/strong). The results show that ScaleErasure achieves a more favorable safety–efficiency trade-off than SLD-medium/strong. Although our method introduces two additional forward passes, the practical overhead remains limited because guidance is applied only at intermediate scales and the extra passes are early-stopped. As shown in Figure4, under the adopted setting, the computation increases by only 7.84%, rather than doubling.
>
> ---
>
> **Lastly, we sincerely thank you again for the detailed and constructive comments, which will help us improve the paper. We are happy to clarify any further questions during the discussion period.**

---

> > ### Author Rebuttal · Reviewer_KhEd · 2026-04-04
> >
> > Thank you for the rebuttal. While most of my concerns have been addressed, I encourage the authors to include latency measurements, as FLOP calculations alone may not accurately reflect actual runtime performance. I will raise my rating.

---

> > > ### Author Response · Authors · 2026-04-04
> > >
> > > Dear reviewer KhEd:
> > >
> > > ---
> > >
> > > Thanks for your helpful suggestion about latency measurement. We report the generation latency below on an NVIDIA RTX A6000, which shows that the early-stop strategy effectively reduces the overhead and that the runtime increase of our method remains acceptable in practice.
> > >
> > > |                                  | Latency (s/sample) |
> > > | :------------------------------- | :----------------: |
> > > | *Base Model*                     |       *2.33*       |
> > > | ScaleErasure (w/o early stop)    |   3.15 (+35.19%)   |
> > > | **ScaleErasure (w/ early stop)** | **2.54 (+9.01%)**  |
> > >
> > > ---
> > >
> > > Thank you sincerely for taking the time to review our rebuttal and for thoughtfully considering our clarifications. We truly appreciate your increased score and your constructive comments, which help us strengthen the quality and clarity of our work.
> > >
> > > Best regards,
> > >
> > > The Authors

---

### Official Review · Reviewer_zASo · 2026-03-16

**Soundness:** 3
**Presentation:** 3
**Significance:** 3
**Originality:** 3
**Overall Recommendation:** 4
**Confidence:** 3

**Summary:**

This paper studies concept erasure for next-scale autoregressive image generation models. The authors propose ScaleErasure, an inference-time method that selectively modifies logits related to unsafe concepts across scales, tokens, and bit channels. Experiments show that the method can effectively remove unsafe concepts while largely preserving overall generation quality.

**Compliance With Llm Reviewing Policy:**

Affirmed.

**Final Justification:**

The paper is quite significant, as it proposes corresponding solutions for safety-related issues and is strongly motivated. In addition, the authors responded constructively to the concerns I raised, such as the addition of other concepts and the user study. Therefore, taking into account the paper’s strengths, weaknesses, and the authors’ rebuttal, I would give this paper a weak accept.

**Key Questions For Authors:**

See weakness.

**Limitations:**

See weakness.

**Strengths And Weaknesses:**

Strength:

1. The paper studies concept erasure for next-scale autoregressive image generation models, a setting that has been less explored compared to diffusion models. It clearly identifies the challenge of semantic entanglement at low-resolution scales and proposes a solution tailored to this generation paradigm.
2. The experiments include both nudity and copyright concept erasure, and show that the method balances effective erasure with generation quality.

Weaknesses:

1. The experiments mainly focus on nudity and a few copyrighted characters. I wonder whether the method has been tested on other types of unsafe concepts (e.g., violence, harmful symbols, or other sensitive categories). It would be helpful to understand how well the approach generalizes to a broader range of concepts.
2. The paper relies entirely on automated metrics for evaluation. It would be helpful to include a small user study to better assess whether the edited images remain visually natural after concept erasure.

---

> ### Author Rebuttal · Authors · 2026-03-31
>
> Dear reviewer zASo:
>
> **We sincerely appreciate your recognition of the motivation and experimental results of our work. We will address your concerns below.**
>
> ---
>
> **`W1:` Limited generalization evaluation across concepts**
>
> **`Response:`**
>
> Thanks for your insightful suggestion. We further conduct experiments on a broader range of concepts, including **Snoopy** and **Mickey Mouse** for copyrighted characters, **Rembrandt** and **Andy Warhol** for art styles, as well as six broader unsafe categories. Specifically, for the six broader unsafe categories, we use a discriminative classifier to measure the probability that a generated image belongs to the corresponding unsafe category.
>
> |$H_a$|Snoopy|Mickey Mouse|Rembrandt|Andy Warhol|
> |-|:-:|:-:|:-:|:-:|
> |UCE|2.34|**8.20**|4.28|4.59|
> |RECE|3.68|2.00|3.75|2.06|
> |ESD-x|**11.00**|3.39|5.05|4.05|
> |ESD-u|7.32|2.43|0.05|4.04|
> |SLD-medium|2.55|-1.74|4.29|1.87|
> |SLD-strong|2.95|-1.47|3.67|2.77|
> |**ScaleErasure (Ours)**|10.68|7.13|**8.23**|**8.26**|
>
> | |Hate|Harassment|Violence|Self-Harm|Shocking|Illegal|*Avg.*|
> |-|:-:|:-:|:-:|:-:|:-:|:-:|:-:|
> |ESD-x|18.18|12.38|23.02|20.97|25.82|16.64|19.5|
> |ESD-u|21.65|16.75|24.87|25.09|33.29|22.01|23.94|
> |UCE|26.41|25.24|33.99|28.71|39.02|23.93|29.55|
> |RECE|23.38|21.48|32.14|28.34|38.79|26.00|28.35|
> |SLD-medium|20.78|15.53|22.09|23.85|29.09|18.98|21.72|
> |SLD-strong|17.32|14.56|21.96|20.22|26.87|**14.03**|19.16|
> |**ScaleErasure (Ours)**|**14.29**|**9.71**|**11.90**|**18.85**|**22.66**|15.96|**15.56**|
>
> These results show that **ScaleErasure generalizes well to a broader range of concepts**. We will include the detailed results in the revised version.
>
> ---
>
> **`W2:` Missing user study**
>
> **`Response:`**
>
> Thanks for your helpful suggestion. We conduct a user study with 10 human evaluators. We randomly sample and shuffle 20 groups of results spanning nudity, copyrighted characters, art styles, and violence. For each group, the evaluators compare the images generated by concept erasure methods against those from the base model, and assign 1–5 scores from three perspectives: **(1) unsafe concept erasure effectiveness**, **(2) preservation of unrelated regions**, and **(3) overall image naturalness**.
>
> | |Total Scores (1)|Total Scores (2)|Total Scores (3)|
> |:-|:-:|:-:|:-:|
> |ESD-x|612|608|621|
> |ESD-u|701|662|676|
> |UCE|646|603|590|
> |RECE|668|628|607|
> |SLD-medium|734|751|748|
> |SLD-strong|759|736|701|
> |**ScaleErasure (Ours)**|**876**|**849**|**795**|
> |*MAX*|*1000*|*1000*|*1000*|
>
> The results further confirm that **ScaleErasure provides the best overall trade-off between erasure effectiveness and image quality**. We will include the detailed results in the revised version.
>
> ---
>
> **Lastly, we sincerely thank you again for the detailed and constructive comments, which will help us improve the paper. We are happy to clarify any further questions during the discussion period.**

---

> > ### Author Rebuttal · Reviewer_zASo · 2026-03-31
> >
> > My concerns have been fully addressed. Additionally, after reviewing the other comments and the authors’ rebuttal, I believe the authors have presented a well-defined and strongly motivated piece of work. The problem they tackle is also challenging within this field. Therefore, I will maintain my original score.

---

> > > ### Author Response · Authors · 2026-04-01
> > >
> > > Dear reviewer zASo:
> > >
> > > Thank you sincerely for taking the time to review our rebuttal and for thoughtfully considering our clarifications. We truly appreciate your positive score and your constructive comments, which help us strengthen the quality and clarity of our work.
> > >
> > > Best regards,
> > >
> > > The Authors

---

### Official Review · Reviewer_DPPD · 2026-03-17

**Soundness:** 2
**Presentation:** 3
**Significance:** 2
**Originality:** 2
**Overall Recommendation:** 3
**Confidence:** 4

**Summary:**

This paper studies concept erasure in next-scale autoregressive image generation. The paper argues that, unlike diffusion models, next-scale AR suffers from stronger semantic compression and entanglement at early low-resolution scales, making precise erasure harder. To address this, the authors propose ScaleErasure, an inference-time method that performs two extra forward passes with unsafe and safe concepts, and selectively applies logits guidance across scale, token, and bit-channel dimensions to steer generation away from unsafe concepts.

**Compliance With Llm Reviewing Policy:**

Affirmed.

**Key Questions For Authors:**

1. Please explain the inconsistency in the reported FID values; this directly affects the confidence in the empirical results.
2. Please provide more detail on how the adapted baselines were implemented and tuned, to establish fairness.
3. How stable is the method under different choices of safe and unrelated concept sets?
4. Can the evaluation be extended to more copyrighted characters or more unsafe concept categories?
5. Can the authors provide human evaluation or a more direct analysis of off-target damage?

**Limitations:**

Yes.

**Strengths And Weaknesses:**

Strengths:
1. Safety control and concept erasure for next-scale AR models are still underexplored, so the paper addresses a timely and worthwhile problem with clear practical relevance.
2.The scale/token/bit-channel decomposition and selective safe guidance form a coherent framework, and the overall method is organized in a clear and easy-to-follow way.
3. The paper provides a plausible motivation for why intervening too early or too late in the scale hierarchy can be suboptimal, which makes the design choices fairly intuitive.
4. In Table 1, the method performs well on total nudity count, MS-COCO FID, and average Ha, suggesting a promising balance between erasure effectiveness and generation quality.
5. The paper includes ablations, threshold studies, and token-selection visualizations, which help the reader better understand the contribution of each component.

Weaknesses:
1. All comparisons are against adapted baselines, but the adaptation and tuning details are not sufficiently explained.
2. Copyright erasure is tested on only two characters, and the unsafe/safe/unrelated concepts are manually defined.
3. Nudity erasure relies heavily on NudeNet and copyright erasure relies on CLIP-based metrics, without enough human evaluation or direct off-target damage analysis.
4. Table 1 reports FID 2.91 for ScaleErasure, while later experimental discussion around the penalization sweep suggests values around 52, which weakens confidence in the results.
5. One part says channels with differences below the threshold are selected, while another says increasing the threshold expands the selected set; this is not explained clearly enough.
6. The contribution feels more like a careful combination of existing steering/localization ideas in a new setting than a major conceptual leap.

---

> ### Author Rebuttal · Authors · 2026-03-31
>
> Dear reviewer DPPD:
>
> **Thank you for your constructive comments. We address them below.**
>
> ---
>
> **`W1 & Q2:` Insufficient details on baseline adaptation and tuning**
>
> **`Response:`**
>
> We will include the adaption details in the revised version.
>
> ---
>
> **`W2 & Q4:` Limited generalization evaluation across concepts**
>
> **`Response:`**
>
> We evaluate on additional concepts, including copyrighted characters (**(1) Snoopy**, **(2) Mickey Mouse**), art styles(**(3) Rembrandt**, **(4) Andy Warhol**), and six broader unsafe categories. These results show the strong generalization of our method.
>
> |$H_a$|(1)|(2)|(3)|(4)|
> |-|:-:|:-:|:-:|:-:|
> |UCE|2.34|**8.20**|4.28|4.59|
> |RECE|3.68|2.00|3.75|2.06|
> |ESD-x|**11.00**|3.39|5.05|4.05|
> |ESD-u|7.32|2.43|0.05|4.04|
> |SLD-medium|2.55|-1.74|4.29|1.87|
> |SLD-strong|2.95|-1.47|3.67|2.77|
> |**Ours**|10.68|7.13|**8.23**|**8.26**|
>
> | |Hate|Harassment|Violence|Self-Harm|Shocking|Illegal|Avg.|
> |-|:-:|:-:|:-:|:-:|:-:|:-:|:-:|
> |ESD-x|18.18|12.38|23.02|20.97|25.82|16.64|19.5|
> |ESD-u|21.65|16.75|24.87|25.09|33.29|22.01|23.94|
> |UCE|26.41|25.24|33.99|28.71|39.02|23.93|29.55|
> |RECE|23.38|21.48|32.14|28.34|38.79|26.00|28.35|
> |SLD-medium|20.78|15.53|22.09|23.85|29.09|18.98|21.72|
> |SLD-strong|17.32|14.56|21.96|20.22|26.87|**14.03**|19.16|
> |**Ours**|**14.29**|**9.71**|**11.90**|**18.85**|**22.66**|15.96|**15.56**|
>
> ---
>
> **`W2 & Q3:` Sensitivity to manually defined safe/unrelated concept sets**
>
> **`Response:`**
>
> Manual concept specification follows common practice in prior methods. To assess the sensitivity, we evaluate several alternative safe and unrelated concept sets, and the results show that our method remains stable across different choices.
>
> |NUDITY (Safe)|NudeNet|FID|
> |-|:-:|:-:|
> |"a fully clothed person"|215|2.88|
> |"a person dressed in normal clothes"|192|2.91|
> |"a modestly dressed person"|168|3.26|
>
> |COPYRIGHT (Safe)|$H_a$ (Pikachu)|$H_a$ (SpongeBob)|
> |-|:-:|:-:|
> |"a cute fictional creature"|7.30|9.30|
> |"a generic toy figure"|8.27|8.35|
> |"a simple colored mascot"|9.03|8.51|
>
> |NUDITY (Unrelated)|NudeNet|FID|
> |-|:-:|:-:|
> |"background, scenery, environment, furniture, architecture, landscape"|214|2.95|
> |"scene, background, indoor, outdoor, furniture, building, plants, sky"|129|3.64|
> |"environment, surroundings, room, wall, floor, furniture, architecture"|203|3.24|
>
> |COPYRIGHT (Unrelated)|$H_a$ (Pikachu)|$H_a$ (SpongeBob)|
> |-|:-:|:-:|
> |"background, scenery, environment, furniture, architecture, landscape"|8.55|9.90|
> |"environment, surroundings, room, wall, floor, furniture, architecture"|8.37|9.81|
> |"scene, background, indoor, outdoor, furniture, building, plants, sky"|8.50|10.16|
>
> ---
>
> **`W3 & Q5:` Missing human evaluation**
>
> **`Response:`**
>
> We conduct a user study with 10 evaluators on 20 shuffled groups covering various concepts. Evaluators compare each erasure method against the base model and assign 1–5 scores for **(1) erasure effectiveness** and **(2)preservation of unrelated regions**. The results show that our method achieves the best overall trade-off.
>
> |Scores|(1)|(2)|
> |-|:-:|:-:|
> |ESD-x|612|608|
> |ESD-u|701|662|
> |UCE|646|603|
> |RECE|668|628|
> |SLD-medium|734|751|
> |SLD-strong|759|736|
> |**Ours**|**876**|**849**|
>
> ---
>
> **`W4 & Q1:` Inconsistency in reported FIDs**
>
> **`Response:`**
>
> As stated in Sec. 5.1, the FID difference comes from different sample sizes: 10,000 for the main results and 500 for hyperparameter search. As shown below, the re-evaluated results of Fig. 7(b)(c) with 10,000 samples are linearly correlated with the original ones, indicating unchanged trends and conclusions.
>
> |$\Delta\tau_\mathrm{token}$|FID|
> |:-:|:-:|
> |-2|4.56|
> |-1.5|3.87|
> |-1|3.40|
> |-0.5|3.11|
> |0|2.91|
> |0.5|2.79|
> |1|2.68|
> |1.5|2.60|
>
> |$\tau_\mathrm{bit}$|FID|
> |:-:|:-:|
> |0.05|2.67|
> |0.1|2.91|
> |0.15|3.08|
> |0.2|3.11|
> |0.25|3.14|
> |0.3|3.14|
> |0.35|3.15|
> |0.4|3.15|
>
> ---
>
> **`W5:` Inconsistency in the explanation of threshold-based channel selection**
>
> **`Response:`**
>
> In Sec. 4.4, we define the selected channels as those whose logits differences are below a predefined threshold. Under this definition, when a larger threshold is used, the selected set naturally becomes larger, since more channels satisfy the inequality.
>
> ---
>
> **`W6:` Incremental novelty of the technical contribution**
>
> **`Response:`**
>
> Our method is not a direct adaptation of existing methods.
>
> (1) **Steering:** Safe Guidance uses only unconditioned, safe, and unsafe forward passes, without the conditional pass. Since the selected locations are already identified as unsafe, the goal is to steer them from the unsafe concept toward the safe one.
>
> (2) **Localization:** Our competition-based localization contrasts unsafe and unrelated concepts. Since unsafe concepts can be entangled with other semantics, contrasting them with unrelated concepts helps isolate the unsafe regions.
>
> ---
>
> **Thank you again for the helpful comments.**

---

### Decision · Program_Chairs · 2026-04-30

**Decision:**

Accept (regular)

**Comment:**

ScaleErasure proposes an inference-time concept erasure method for next-scale autoregressive (AR) image generation. Two additional forward passes (conditioned on unsafe and safe concepts) produce guidance that is selectively applied across three dimensions — scale, token, and bit channel — to suppress unsafe content while preserving unrelated semantics.

**Strength**
- Timely problem: concept erasure in next-scale AR generation is largely unexplored, and the scale/token/bit-channel decomposition is a coherent, well-motivated framework tailored to the semantic entanglement at early scales.
- Empirically strong on Infinity-2B across nudity and copyrighted-character erasure; ablations and threshold studies support each design component.
- Inference-time only, with ~9% latency overhead under the early-stop strategy, making it practical to deploy.
- Rebuttal substantially broadened evaluation (additional copyrighted characters, art styles, six unsafe categories, Infinity-8B generalization, 10-evaluator user study) and resolved the FID inconsistency.

**Weakness**
- Novelty is incremental: the method is a careful combination of existing steering + localization ideas adapted to a new setting, rather than a major conceptual leap.
- Token-level threshold still requires light per-setting tuning; behavior under adversarial prompts / more diverse base models remains lightly explored.
- One reviewer noted that prompt-rewriting alternatives were not empirically compared, though authors argued convincingly that prompt rewriting cannot address model-internal generative dynamics.

Overall, a well-executed, deployable solution in an underexplored setting; three of four reviewers converged on weak accept post-rebuttal, and remaining concerns are about scope rather than soundness.